# Loss function based second-order Jensen inequality and its application to particle variational inference

**Futoshi Futami, Tomoharu Iwata, Naonori Ueda**
NTT Communication Science Laboratories
`{futoshi.futami.uk,tomoharu.iwata.gy,naonori.ueda.fr}@hco.ntt.co.jp`

**Issei Sato, Masashi Sugiyama**
The University of Tokyo
`sato@g.ecc.u-tokyo.ac.jp,sugi@k.u-tokyo.ac.jp`

## Abstract

Bayesian model averaging, obtained as the expectation of a likelihood function by a posterior distribution, has been widely used for prediction, evaluation of uncertainty, and model selection. Various approaches have been developed to efficiently capture the information in the posterior distribution; one such approach is the optimization of a set of models simultaneously with interaction to ensure the diversity of the individual models in the same way as ensemble learning. A representative approach is particle variational inference (PVI), which uses an ensemble of models as an empirical approximation for the posterior distribution. PVI iteratively updates each model with a repulsion force to ensure the diversity of the optimized models. However, despite its promising performance, a theoretical understanding of this repulsion and its association with the generalization ability remains unclear. In this paper, we tackle this problem in light of PAC-Bayesian analysis. First, we provide a new second-order Jensen inequality, which has the repulsion term based on the loss function. Thanks to the repulsion term, it is tighter than the standard Jensen inequality. Then, we derive a novel generalization error bound and show that it can be reduced by enhancing the diversity of models. Finally, we derive a new PVI that optimizes the generalization error bound directly. Numerical experiments demonstrate that the performance of the proposed PVI compares favorably with existing methods in the experiment.

## 1 Introduction

Bayesian model averaging (BMA) has been widely employed for prediction, evaluation of uncertainty, and model selection in Bayesian inference. BMA is obtained as the expectation of a likelihood function by a posterior distribution and thus it contains information of each model drawn from the posterior distribution [21]. Since estimating the posterior distribution is computationally difficult in practice, various approximations have been developed to efficiently capture the diversity in the posterior distribution [21, 1, 2].

One of these recently proposed approaches involves optimizing a set of models simultaneously with interaction to ensure the diversity of the individual models, similar to ensemble learning. One notable example is particle variational inference (PVI) [17, 28], which uses an ensemble as an empirical approximation for the posterior distribution. Such PVI methods have been widely employed in variational inference owing to their high computational efficiency and flexibility. They iteratively update the individual models and the update equations contain the gradient of the likelihood function and the repulsion force that disperses the individual models. Thanks to this repulsion term, the obtained ensemble can appropriately approximate the posterior distribution. When only one model is

used in PVI, the update equation is equivalent to that of the maximum a posteriori (MAP) estimation. Other methods have been developed apart from PVI, especially for latent variable models, which have introduced regularization to the MAP objective function to enforce the diversity in the ensemble. A notable example of such methods is the determinantal point process (DPP) [29].

Despite successful performances of these methods in practice [22, 18, 17, 28, 5, 27], a theoretical understanding of the repulsion forces remains unclear. Some previous studies considered PVI as a gradient flow in Wasserstein space with an infinite ensemble size [16, 12] and derived the convergence theory. However, an infinite ensemble size is not a practical assumption and no research has been conducted to analyze the repulsion force related to the generalization.

BMA can be regarded as a special type of ensemble learning [26], and recent work has analyzed the diversity of models in ensemble learning in light of the PAC-Bayesian theory [19]. They reported that the generalization error is reduced by increasing the variance of *the predictive distribution*. However, the existing posterior approximation methods, such as PVI and DPPs, enhance the diversity with the repulsion of the parameters or models rather than the repulsion of the predictive distribution. We also found that the analysis in a previous work [19] cannot be directly extended to the repulsion of parameters or loss functions (see Appendix F). In addition, when the variance of the predictive distribution is included in the objective function in the variational inference, the obtained model shows large epistemic uncertainty, which hampers the fitting of each model to the data (see Section 5).

Based on these findings, this study aims to develop a theory that explains the repulsion forces in PVI and DPPs and elucidates the association of the repulsion forces with the generalization error. To address this, we derive the novel second-order Jensen inequality and connect it to the PAC-Bayesian generalization error analysis. Our second-order Jensen inequality includes the information of the variance of loss functions. Thanks to the variance term, our bound is tighter than the standard Jensen inequality. Then, we derive a generalization error bound that includes the repulsion term, which means that enhancing the diversity is necessary to reduce the generalization error. We also show that PVI and DPPs can be derived from our second-order Jensen inequality, and indicate that these methods work well from the perspective of the generalization error. However, since these existing methods do not minimize the generalization error upper bound, there is still room for improvement. In this paper, we propose a new PVI that directly minimize the generalization error upper bound and empirically demonstrate its effectiveness.

Our contributions are summarized as follows:

1. We derive a novel second-order Jensen inequality that inherently includes the variance of loss functions. Thanks to this variance term, our second-order Jensen inequality is tighter than the standard Jensen inequality. We then show that enhancing the diversity is important for reducing the generalization error bound in light of PAC-Bayesian analysis.

2. From our second-order Jensen inequality, we derive the existing PVI and DPPs. We demonstrate that these methods work well even at a finite ensemble size, since their objective functions includes valid diversity enhancing terms to reduce the generalization error.

3. We propose a new PVI that minimizes the generalization error bound directly. We numerically demonstrate that the performance of our PVI compares favorably with existing methods.

## 2 Background

In this section, we briefly review PVI, DPPs, and PAC-Bayesian analysis.

### 2.1 Particle variational inference

Assume that training dataset $\mathcal{D} = (x_1, \ldots, x_D)$ is drawn independently from unknown data generating distribution $\nu(x)$. Our goal is to model $\nu(x)$ by using a parametrized model $\ln p(x|\theta)$, where $\theta \in \Theta \subset \mathbb{R}^d$. We express $p(\mathcal{D}|\theta) = \sum_{d=1}^{D} \ln p(x_d|\theta)$ and assume a probability distribution over parameters. In Bayesian inference, we incorporate our prior knowledge or assumptions into a prior distribution $\pi(\theta)$. This is updated to a posterior distribution $p(\theta|\mathcal{D}) \propto p(\mathcal{D}|\theta)\pi(\theta)$, which incorporates the observation $\mathcal{D}$. Let us consider the approximation of $p(\theta|\mathcal{D})$ with $q(\theta)$. We predict a new data point by a predictive distribution $p(x) = \mathbb{E}_{q(\theta)} p(x|\theta)$, where $\mathbb{E}$ denotes the expectation. This expectation over the posterior is often called BMA [21].

Table 1: Particle variational inference methods. $I$ is an $N \times N$ identity matrix and $K_{i,j} := K(\theta_i, \theta_j)$.

| Methods | $v(\theta)$ |
|---------|-------------|
| SVGD[17] | $\frac{1}{N}\sum_{j=1}^{N} K_{ij}\partial_{\theta_j}\log p(\mathcal{D}|\theta_j)\pi(\theta_j) + \partial_{\theta_j}K_{ij}$ |
| w-SGLD[3] | $\partial_{\theta_i}\log p(\mathcal{D}|\theta_i)\pi(\theta_i) + \sum_{j=1}^{N}\frac{\partial_{\theta_j}K_{ij}}{\sum_{k=1}^{N}K_{jk}} + \frac{\sum_{j=1}^{N}\partial_{\theta_j}K_{ji}}{\sum_{k=1}^{N}K_{ik}}$ |
| GFSD[15] | Without the second term in w-SGLD |
| GFSF[15] | $\partial_{\theta_i}\log p(\mathcal{D}|\theta_i)\pi(\theta_i) + \frac{1}{N}\sum_{j=1}^{N}((K+cI)^{-1})_{ij}\partial_{\theta_j}K_{ij}$ |

Assume that we draw $N$ models from the posterior distribution and calculate BMA. We denote those drawn models as an empirical distribution $\rho_{\mathrm{E}}(\theta) = \frac{1}{N}\sum_{i=1}^{N}\delta_{\theta_i}(\theta)$, where $\delta_{\theta_i}(\theta)$ is the Dirac distribution that has a mass at $\theta_i$. We also refer to these $N$ models as particles. The simplest approach to obtain these $N$ particles is MAP estimate that updates parameters independently with gradient descent (GD) as follows [28]:

$$\theta_i^{\mathrm{new}} \leftarrow \theta_i^{\mathrm{old}} + \eta\partial_\theta \ln p(D|\theta_i^{\mathrm{old}})\pi(\theta_i^{\mathrm{old}}), \tag{1}$$

where $\eta \in \mathbb{R}^+$ is a step size. In BMA, we are often interested in the multi-modal information of the posterior distribution. In such a case, MAP estimate is not sufficient because $N$ optimized particles do not necessarily capture the appropriate diversity of the posterior distribution. Instead, particle variational inference (PVI) [17, 28] approximates the posterior through iteratively updating the empirical distribution by interacting them with each other:

$$\theta_i^{\mathrm{new}} \leftarrow \theta_i^{\mathrm{old}} + \eta v(\{\theta_{i'}^{\mathrm{old}}\}_{i'=1}^{N}), \tag{2}$$

where $v(\{\theta\})$ is the update direction and explicit expressions are summarized in Table 1. Basically, $v$ is composed of the gradient term and the repulsion term. In Table 1, the repulsion terms contain the derivative of the kernel function $K$, and the Gaussian kernel [23] is commonly used. When the bandwidth of $K$ is $h$, the repulsion term is expressed as $\partial_{\theta_i}K(\theta_i, \theta_j) = -h^{-2}(\theta_i - \theta_j)e^{-(2h^2)^{-1}\|\theta_i - \theta_j\|^2}$, where $\|\cdot\|$ denotes the Euclidean norm. We refer to this as a parameter repulsion. Note that the repulsion term depends on the distance between particles, and the closer they are, the stronger force is applied. This moves $\theta_i$ away from $\theta_j$, and thus particles tend not to collapse to a single mode.

For over-parametrized models such as neural networks, since the repulsion in the parameter space is not enough for enhancing the diversity, function space repulsion force for supervised tasks was developed [28]. We call it function space PVI (f-PVI). Pairs of input-output data are expressed as $\mathcal{D} = \{(x_d, y_d)\}_{d=1}^{D}$. We consider the model $p(y|x, \theta) = p(y|f(x;\theta))$ where $f(x;\theta)$ is a $c$-dimensional output function parametrized by $\theta$ and $x$ is an input. Furthermore, we consider the distribution over $f$ and approximate it by a size-$N$ ensemble of $f$, which means that we prepare $N$ parameters (particles) $\{\theta_i\}_{i=1}^{N}$. We define $f_i(x) := f(x;\theta_i)$. When we input the minibatch with size $b$ into the model, we express it as $f_i(\boldsymbol{x}_{1:b}) = (f_i(x_1), \ldots, f_i(x_b)) \in \mathbb{R}^{cb}$. Then the update equation is given as

$$\theta_i^{\mathrm{new}} \leftarrow \theta_i^{\mathrm{old}} + \eta\frac{\partial f_i(\boldsymbol{x}_{1:b})}{\partial\theta_i}\bigg|_{\theta_i=\theta_i^{\mathrm{old}}} v(\{f_i(\boldsymbol{x}_{1:b})\}_{i=1}^{N}), \tag{3}$$

where $v(\{f_i\})$ is obtained by replacing $\ln p(\mathcal{D}|\theta)\pi(\theta)$ with $(D/b)\sum_{d=1}^{b}\ln p(y_d|f_i(x_d))\pi(f_i)$, where $\pi(f)$ is a prior distribution over $f$ and the Gram matrix $K(\theta_i, \theta_j)$ is replaced with $K(f_i^b, f_j^b)$ in Table 1. See appendix C.1 for details. Then, f-PVI modifies the loss signal so that models are diverse. We refer to the repulsion term of f-PVI as a model repulsion. We express $f_i^b := f_i(\boldsymbol{x}_{1:b})$ for simplicity. When we use the Gaussian kernel, the model repulsion is expressed as

$$\partial_{\theta_i}K(f_i^b, f_j^b) = -h^{-2}(f_i^b - f_j^b)e^{-\|f_i^b - f_j^b\|^2/(2h^2)}\partial_{\theta_i}f_i^b. \tag{4}$$

Thus, the model repulsion pushes model $f_i$ away from $f_j$.

## 2.2 Regularization based methods and determinantal point processes

Another common approach for enhancing the diversity for latent variable models is based on regularization. A famous example is the determinantal point process (DPP) [29], in which we maximize

$$\mathbb{E}_{\rho_{\mathrm{E}}}\ln p(\mathcal{D}|\theta)\pi(\theta) + \ln\det K, \tag{5}$$

where $K$ is the kernel Gram matrix defined by $K_{i,j} = K(\theta_i, \theta_j)$. This log-determinant term is essentially a repulsion term that enhances the diversity in the parameter space.

## 2.3 PAC-Bayesian theory

Here, we introduce PAC-Bayesian theory [7]. We define the generalization error as the cross-entropy:

$$\text{CE} := \mathbb{E}_{\nu(x)}[-\ln \mathbb{E}_{q(\theta)} p(x|\theta)], \tag{6}$$

which corresponds to the Kullback-Leibler (KL) divergence. Our goal is to find $q(\theta)$ that minimizes the above CE. In many Bayesian settings, we often minimize not CE but a surrogate loss [19] that is obtained by the Jensen inequality:

$$\mathbb{E}_{\nu(x)}[-\ln \mathbb{E}_{q(\theta)} p(x|\theta)] \leq \mathbb{E}_{\nu(x), q(\theta)}[-\ln p(x|\theta)]. \tag{7}$$

Since the data generating distribution is unknown, we approximate it with a training dataset as $\mathbb{E}_{\nu(x), q(\theta)}[-\ln p(x|\theta)] \approx \mathbb{E}_{q(\theta)} \frac{1}{D} \sum_{d=1}^{D} [-\ln p(x_d|\theta)]$. The PAC-Bayesian generalization error analysis provides the probabilistic relation for this approximation as follows:

**Theorem 1** (Germain [7]). *For any prior distribution $\pi$ over $\Theta$ independent of $\mathcal{D}$ and for any $\xi \in (0, 1)$ and $c > 0$, with probability at least $1 - \xi$ over the choice of training data $\mathcal{D} \sim \nu^{\otimes D}(x)$, for all probability distributions $q$ over $\Theta$, we have*

$$\mathbb{E}_{\nu(x), q(\theta)}[-\ln p(x|\theta)] \leq \mathbb{E}_{q(\theta)} \frac{1}{D} \sum_{d=1}^{D} [-\ln p(x_d|\theta)] + \frac{\text{KL}(q, \pi) + \ln \xi^{-1} + \Psi_{\pi, \nu}(c, D)}{cD}, \tag{8}$$

*where $\Psi_{\pi, \nu}(c, D) := \ln \mathbb{E}_{\pi} \mathbb{E}_{\mathcal{D} \sim \nu^{\otimes D}(x)} \exp[cD(-\mathbb{E}_{\nu(x)} \ln p(x|\theta) + D^{-1} \sum_{d=1}^{D} \ln p(x_d|\theta))]$.*

The Bayesian posterior is the minimizer of the right-hand side of the PAC-Bayesian bound when $c = 1$. Recently, the PAC-Bayesian bound has been extended so that it includes the diversity term [19]. Under the same assumptions as Theorem 1 and for all $x, \theta, p(x|\theta) < \infty$, we have

$$\text{CE} \leq -\mathbb{E}_{\nu, q}[\ln p(x|\theta) + V(x)] \leq -\mathbb{E}_q \frac{1}{D} \sum_{d=1}^{D} [\ln p(x_d|\theta) + V(x_d)] + \frac{\text{KL}(q, \pi) + \frac{\ln \xi^{-1} + \Psi'_{\pi, \nu}(c, D)}{2}}{cD}, \tag{9}$$

where $V(x) := (2 \max_\theta p(x|\theta)^2)^{-1} \mathbb{E}_{q(\theta)} \left[ (p(x|\theta) - \mathbb{E}_{q(\theta)} p(x|\theta))^2 \right]$ is the variance of the predictive distribution and $\Psi'_{\pi, \nu}(c, D)$ is the modified constant of $\Psi_{\pi, \nu}(c, D)$ (see Appendix C.2 for details). A similar bound for ensemble learning, that is, $q(\theta)$ as an empirical distribution, was also previously proposed [19] (see Appendix C.3). This bound was derived directly from the second-order Jensen inequality derived in another work [14]. Furthermore, the diversity comes from the variance of the predictive distribution, which is different from PVI and DPPs because their repulsion is in the parameter or model space. Note that we cannot directly change the variance of the predictive distribution to that of PVI or DPPs because it requires an inequality that is contrary to the Jensen inequality. We also found that directly optimizing the upper bound of Eq.(9), referred to as $\text{PAC}_{\text{E}}^2$, results in a too large variance of the predictive distribution which is too pessimistic for supervised learning tasks (see Section 5).

## 3 Method

Here, we derive our novel second-order Jensen inequality based on the variance of loss functions and then derive a generalization error bound. Then, we connect our theory with existing PVI and DPPs.

### 3.1 A novel second-order Jensen inequality

First, we show the second-order equality, from which we derive our second-order Jensen inequality.

**Theorem 2.** *Let $\psi$ be a twice differentiable monotonically increasing concave function on $\mathbb{R}^+$, $Z$ be a random variable on $\mathbb{R}^+$ that satisfies $\mathbb{E}[Z^2] < \infty$, and its probability density be $p_Z(z)$. Define a constant $\mu := \psi^{-1}(\mathbb{E}[\psi(Z)])$. Then, we have*

$$\mathbb{E}[Z] = \mu - \left( 2 \frac{d\psi(\mu)}{dz} \right)^{-1} \int_{\mathbb{R}^+} \left[ \frac{d^2 \psi(c(z))}{dz^2} (z - \mu)^2 \right] p_Z(z) dz, \tag{10}$$

*where $c(z)$ is a constant between $z$ and $\mu$ that is defined from the Taylor expansion (see Appendix D.1 for details).*

*Proof sketch:* There exists a constant $c(z)$ between $z$ and $\mu$ s.t. $\psi(z) = \psi(\mu) + \frac{d\psi(\mu)}{dz}(z-\mu) + \frac{1}{2}\frac{d^2\psi(c(z))}{dz^2}(z-\mu)^2$ from the Taylor expansion. Then we take the expectation. Full proof is given in Appendix D.1. $\square$

This theorem states the deviation of $\mathbb{E}Z$ from $\mu$ when $\psi$ is applied to $Z$. By setting $\psi(\cdot) = \ln(\cdot)$ and $Z = p(x|\theta)$ and applying $\ln$ to both hand sides of Eq.(10), we have the following equality:

**Corollary 1.** *If for all $x$ and $\theta$, $p(x|\theta) < \infty$, we have*

$$\mathbb{E}_{q(\theta)} \ln p(x|\theta) = \ln \mathbb{E}_{q(\theta)} p(x|\theta) - \ln\left(1 + \mathbb{E}_{q(\theta)}(2g(\theta,x)^2)^{-1}(e^{\ln p(x|\theta)} - e^{\mathbb{E}_{q(\theta)} \ln p(x|\theta)})^2\right), \quad (11)$$

*where $g(\theta, x)$ is a constant between $p(x|\theta)$ and $e^{\mathbb{E}_{q(\theta)} \ln p(x|\theta)}$ that is defined from the Taylor expansion (see Appendix D.2 for details).*

**Remark 1.** *Recall that the standard Jensen inequality is $\mathbb{E}_q \ln p(x|\theta) \leq \ln \mathbb{E}_q p(x|\theta)$, and its gap is called the Jensen gap. In Eq.(11), the second term of the right-hand side is always positive. Thus, this term corresponds to the Jensen gap. Remarkably when we use the standard Jensen inequality, this information is lost. We clarify the meaning of this term below. Also note that our second-order equality is different from those of the previous works [19, 14] (see Appendix F for details).*

Next, we show our first main result, loss function based second-order Jensen inequality:

**Theorem 3.** *Under the same assumption as Corollary 1,*

$$\mathbb{E}_{q(\theta)} \ln p(x|\theta) \leq \ln \mathbb{E}_{q(\theta)} p(x|\theta) - \underbrace{\mathbb{E}_{q(\theta)}\left(\frac{\ln p(x|\theta) - \mathbb{E}_{q(\theta)} \ln p(x|\theta)}{2h(x,\theta)}\right)^2}_{:= \mathrm{R}(x,h)}, \quad (12)$$

*where*
$$h(x,\theta)^{-2} = \exp\left(\ln p(x|\theta) + \mathbb{E}_{q(\theta)} \ln p(x|\theta) - 2\max_\theta \ln p(x|\theta)\right). \quad (13)$$

*Proof sketch:* Apply $\sqrt{\alpha\beta} \leq \frac{\alpha-\beta}{\ln\alpha-\ln\beta}$ for any $\alpha, \beta > 0$ to Eq.(11). Full proof is given in Appendix D.3. $\square$

**Remark 2.** *$R$ is the weighted variance of loss functions, and it is always positive. Thus, this inequality is always tighter than the Jensen inequality, and the equality holds if the weighted variance is zero. Compared to the results of the previous works [19, 14] that used the predictive variance in the inequality, our bound focuses on the variance of loss functions. We refer to our repulsion term $R$ as a loss repulsion.*

Then, by rearranging Eq.(12) and taking the expectation, we have the following inequality:

$$\mathrm{CE} \leq -\mathbb{E}_{q(\theta),\nu(x)}[\ln p(x|\theta)] - \mathbb{E}_{\nu(x)}\mathrm{R}(x,h) \leq -\mathbb{E}_{q(\theta),\nu(x)}[\ln p(x|\theta)]. \quad (14)$$

Using this inequality, we obtain the second-order PAC-Bayesian generalization error bound:

**Theorem 4.** *(See Appendix D.4 for the complete statement) Under the same notation and assumptions as Theorems 1 and 3, with probability at least $1 - \xi$, we have*

$$\mathrm{CE} \leq -\mathbb{E}_{\nu,q}[\ln p(x|\theta) + R(x,h)] \leq -\mathbb{E}_q \frac{1}{D}\sum_{d=1}^{D}[\ln p(x_d|\theta) + R(x_d,h_m)] + \frac{\mathrm{KL}(q,\pi) + \frac{\ln\xi^{-1} + \Psi''_{\pi,\nu}(c,D)}{3}}{cD}, \quad (15)$$

*where $\Psi''_{\pi,\nu}(c, D)$ is the modified constant of $\Psi_{\pi,\nu}(c, D)$ and $R(x,h_m)$ is $R(x,h)$ in Eq.(12) replacing $h(x,\theta)^{-2}$ of Eq.(13) with $h_m(x,\theta)^{-2} = \exp(\ln p(x|\theta) + \min_\theta \ln p(x|\theta) - 2\max_\theta \ln p(x|\theta))$.*

*Proof sketch.* We express $\mathbb{E}_{q(\theta)}[\ln p(x|\theta) + R(x,h_m)]$ as $\mathbb{E}_{q(\theta)q(\theta')q(\theta'')}L(x,\theta,\theta',\theta'')$ where $L(x,\theta,\theta',\theta'') := \ln p(x|\theta) + (2h_m(x,\theta))^{-2}(\ln p(x|\theta)^2 - 2\ln p(x|\theta)\ln p(x|\theta') + \ln p(x|\theta')\ln p(x|\theta''))$. Then, we apply the same proof technique as Theorem 1 [7] to the loss function $L(x,\theta,\theta',\theta'')$ with $\lambda = 3cD$. Full proof is given in Appendix D.4. $\square$

**Remark 3.** *To reduce the upper bound of the generalization error, Eq.(15), we need to control the trade-off between the data fitting term of the negative log-likelihood and enhancing the diversity of the models based on the loss repulsion term $R$.*

**Remark 4.** *In the definition of $h_m$, $\min_\theta \ln p(x|\theta)$ is too pessimistic in some cases. If we additionally assume that there exists a positive constant $M$ s.t. $\mathbb{E}_{q(\theta)}[\ln p(x|\theta)]^2 < M < \infty$, we can replace $\min_\theta \ln p(x|\theta)$ with $\mathrm{Median}_\theta(\ln p(x|\theta)) - M^{1/2}$ in $h_m$ in Theorem 4 (see Appendix D.4.1).*

Compared to Eq.(9), our bound focuses on the variance of loss functions, which has a direct connection to the repulsion of PVI and DPPs (see Section 3.2). Furthermore, we show that optimizing the upper bound in Eq.(15) shows competitive performance with the existing state-of-the-art PVI (see Section 5). Note that this inequality is not restricted to the case where $q(\theta)$ is an empirical distribution. We can also use this for parametric variational inference [21].

## 3.2 Diversity in ensemble learning and connection to existing methods

In the following, we focus on the ensemble setting and use a finite set of parameters as $\rho_E(\theta) := \frac{1}{N} \sum_{i=1}^{N} \delta_{\theta_i}(\theta)$ and discuss the relationship of our theory and existing methods. We show the summary of the relationships in Appendix J.

### 3.2.1 Covariance form of the loss repulsion

To emphasize the repulsion between models, we upper-bound Eq.(12) using the covariance:

**Theorem 5.** *Under the same assumption as Corollary 1, we have*

$$\mathbb{E}_{\rho_E(\theta)} \ln p(x|\theta) \leq \ln \mathbb{E}_{\rho_E(\theta)} p(x|\theta) - \frac{1}{2(2h_w(x,\theta))^2 N^2} \sum_{i,j=1}^{N} \left( \ln p(x|\theta_i) - \ln p(x|\theta_j) \right)^2, \quad (16)$$

*where $h_w(x,\theta)^{-2} = \exp\left( \min_i \ln p(x|\theta_i) + \frac{1}{N} \sum_{i=1} \ln p(x|\theta_i) - 2\max_j \ln p(x|\theta_j) \right)$.*

See Appendix D.5 for the proof. We can also show a generalization error bound like Theorem 4 for the ensemble learning setting (see Appendix E.1). In existing PVI and DPPs, the repulsion is based not on the loss function but the parameters or models, as seen in Section 2. We derive the direct connection between our loss repulsion and the model and parameter repulsion below.

### 3.2.2 Relation to w-SGLD and model repulsion

First, from Eq.(16), we derive the direct connection to w-SGLD, which is an instance of PVI introduced in Section 2. Let us define an $N \times N$ kernel Gram matrix $G$ whose $(i,j)$ element is defined as

$$G_{ij} := \exp\left( -(8h_w^2)^{-1} \left( \ln p(x|\theta_i) - \ln p(x|\theta_j) \right)^2 \right). \quad (17)$$

Applying the Jensen inequality to Eq.(16), we obtain

$$\ln \mathbb{E}_{\rho_E(\theta)} p(x|\theta) \geq \mathbb{E}_{\rho_E(\theta)} \ln p(x|\theta) - \frac{1}{N} \sum_{i=1}^{N} \ln \sum_{j=1}^{N} \frac{G_{ij}}{N} \geq \mathbb{E}_{\rho_E(\theta)} \ln p(x|\theta). \quad (18)$$

This is tighter than the standard Jensen inequality. To derive the relation to w-SGLD, we optimize the middle part of Eq.(18) by gradient descent. We express $L(\theta_i) := \ln p(x|\theta_i)$ and do not consider the dependency of $h_w$ on $\theta$ for simplicity. By taking the partial derivative with respect to $\theta_i$, we have

$$\partial_{\theta_i} \ln p(x|\theta_i) + \left( \sum_{j=1}^{N} \frac{\partial_{L(\theta_j)} G_{ij}}{\sum_{k=1}^{N} G_{jk}} + \frac{\sum_{j=1}^{N} \partial_{L(\theta_j)} G_{ji}}{\sum_{k=1}^{N} G_{ik}} \right) \partial_{\theta_i} L(\theta_i). \quad (19)$$

The second term corresponds to the repulsion term, which is equivalent to that of w-SGLD shown in Table 1. The difference is that our Gram matrix $G$ in Eq.(19) depends on the loss function rather than the parameter or model. Using the mean value theorem, it is easy to verify that there exists a constant $C$ such that $\| \ln p(x|\theta_i) - \ln p(x|\theta_j) \|^2 = \| C(\theta_i - \theta_j) \|^2$ (see Appendix G for details), thus we can easily transform the loss repulsion to the parameter or model repulsion.

However, since we cannot obtain the explicit expression of the constant $C$, it is difficult to understand the intuitive relation between our loss repulsion and the parameter or model repulsion. Instead, here we directly calculate $\partial_{\theta_i} G_{i,j}$ and discuss the relation. Due to the space limitation, we only show the relation to the model repulsion in the regression task of f-PVI. See Appendix G for the complete statement including the classification setting of f-PVI and the parameter repulsion of PVI. Following the setting in Section 2.1, for a regression problem, we assume that $p(y|f(x;\theta))$ is the

Gaussian distribution with unit variance for simplicity. We define $L(f_i) := \ln p(y|f(x;\theta_i))$ and $dL_{ij} := \partial_{f_i} L(f_i) + \partial_{f_j} L(f_j)$. The derivative of the Gram matrix $G$ is expressed as

$$\partial_{\theta_i} G_{ij} = -(\underbrace{(f_i - f_j)\|dL_{ij}\|^2}_{i)} + \underbrace{\partial_{f_i} L(f_i) dL_{ij}\|f_i - f_j\|^2}_{ii)})(4h_w)^{-2} G_{ij} \partial_{\theta_i} f_i. \qquad (20)$$

The first term $i)$ corresponds to the model repulsion of f-PVI shown in Eq.(30) and the second term $ii)$ is the correction term based on the loss function. Thus, our loss repulsion can be translated to the model repulsion of f-PVI plus the correction term using the loss function.

In conclusion, we have confirmed that w-SGLD is directly related to our theory. For Eq.(18), we can also derive a generalization error bound like Theorem 4 (see Appendix E.2 for details). This means that w-SGLD controls the trade-off between the model fitting and enhancing the diversity by optimizing the generalization error bound. This explains the reason why w-SGLD still works well even with a finite ensemble size.

### 3.2.3 Relation to other PVIs

Here, we derive other PVI and DPPs from our theory. First, we derive GFSF shown in Table 1. We express the identity matrix with size $N$ as $I$. We obtain the following upper bound from Eq.(16):

**Theorem 6.** *Under the same assumption as Corollary 1, for any real value $\epsilon \in (1, N-1)$ and for any positive constant $\tilde{h}$, we have*

$$\ln \mathbb{E}_{\rho_{\mathrm{E}}(\theta)} p(x|\theta) \geq \mathbb{E}_{\rho_{\mathrm{E}}(\theta)} \ln p(x|\theta) - \frac{2}{\tilde{h}N} \ln \det(\epsilon I + K) + \frac{2 \ln N}{\tilde{h}N} \geq \mathbb{E}_{\rho_{\mathrm{E}}(\theta)} \ln p(x|\theta), \quad (21)$$

*where $K$ is an $N \times N$ kernel Gram matrix of which $(i, j)$ element is defined as*

$$K_{ij} := \exp\left(-\tilde{h} \ln N (4h_w)^{-2} \left(\ln p(x|\theta_i) - \ln p(x|\theta_j)\right)^2\right). \qquad (22)$$

The proof is shown in Appendix D.6. This is tighter than the standard Jensen inequality.

**Remark 5.** *Eq.(21) holds for any positive bandwidth $\tilde{h}$. $\tilde{h}$ defines the tightness Eq.(21).*

**Remark 6.** *In Eq.(22), we used the scaling of $\ln N$ to define $K$. This is motivated by the median trick of the existing PVI, which tunes the bandwidth as $\ln N/\mathrm{median}^2$. This scaling implies that, for each $i$, $\sum_j K_{ij} \approx 1 + \frac{1}{N}$ holds. We found that using this scaling is necessary to obtain the bound Eq.(21). We conjecture that this is the reason why scaling the bandwidth is important for PVI in practice.*

In the same way as Eq.(19) for w-SGLD, we also optimize the middle term in Eq.(21) by GD. By taking the partial derivative, we have the following update equation:

$$\partial_{\theta_i} \mathbb{E}_{\rho_{\mathrm{E}}(\theta)} \log p(x|\theta_i) + \frac{2}{\tilde{h}N} \sum_j (K + \epsilon I)_{ij}^{-1} \nabla_{L(\theta_j)} K_{ij} \partial_{\theta_i} L(\theta_i). \qquad (23)$$

See appendix D.7 for the proof. The second term is the repulsion force, which is the same as that in the update equation of GFSF in Table 1.

Next we consider the relation to DPPs. Using the trace inequality [8] to Eq.(16) and using a Gram matrix $\tilde{G}$ whose $(i, j)$ element is $G_{ij}^{\frac{1}{2}}$ in Eq.(17), we obtain

$$\ln \mathbb{E}_{\rho_{\mathrm{E}}} p(x|\theta) \geq \mathbb{E}_{\rho_{\mathrm{E}}} \ln p(x|\theta) + \frac{2}{N} \ln \det \tilde{G} - \ln N \qquad (24)$$

The proof is shown in Appendix D.8. The lower bound term of Eq.(24) is equivalent to the objective function of the DPP introduced in Eq.(5).

For Eqs.(21) and (24), we can derive a PAC-Bayesian generalization error bound like Theorem 4 (see Appendix E.2 for details). Moreover, we can connect our loss repulsion in Eqs.(21) and (24) to model and parameter repulsions in the same way as w-SGLD. Accordingly, GFSF and DPP are closely related to the second-order generalization error bound.

In conclusion, we derived PVI and DPPs from our theory based on the second-order Jensen inequality. On the other hand, we found it difficult to show its relation to SVGD, since it has the kernel smoothing coefficient in the derivative of the log-likelihood function. We leave it for future work.

# 4 Discussion and Related work

In this section, we discuss the relationship between our theory and existing work.

## 4.1 Theoretical analysis of PVI

Originally, PVI was derived as an approximation of the gradient flow in Wasserstein space and its theoretical analysis has been done with an infinite ensemble size [16, 12]. In practice, however, an infinite ensemble size is not realistic and various numerical experiments showed that PVI still works well even with a finite ensemble size [17, 28]. Accordingly, our analysis aimed to clarify why PVI works well with such a finite-size ensembles. On the other hand, as discussed in Section 3.2.2, there is a difference between our loss repulsion and the parameter and model repulsion used in the existing works, and thus it would be an interesting direction to extend our theory to fill such a gap.

## 4.2 Relation to existing second-order Jensen inequality

Recently, some works derived tighter Jensen inequalities [14, 6]. Liao and Berg [14] derived a second-order Jensen inequality, and Gao et al. [6] worked on a higher-order inequality. Masegosa [19] combined the earlier result [14] with the PAC-Bayesian theory. Although our bound is also a second-order Jensen inequality, its derivation and behavior are completely different from them. In Liao [14], their second-order Jensen inequality includes the term of a variance of a random variable, and Masagosa [19] considered $p(x|\theta)$ to be a corresponding random variable that depends on $\theta$. Thus, the second-order inequalities depend on the variance of the predictive distribution $\mathbb{E}_{p(\theta)}[p(x|\theta)]$. On the other hand, our bound is based on our novel second-order equality shown in Theorem 2, which leads to the variance of a loss function as shown in Theorem 3. By using the variance of loss functions, we can directly connect our theory to existing PVI and DPPs as shown in Section 3.2. Moreover, as shown in Section 5, including the predictive variance in the objective function results in a large epistemic uncertainty, which means that individual models do not fit well. On the other hand, ours does not show this phenomenon. Consequently, our result can be regarded as an extension of the earlier work [14, 6, 19] that directly focuses on a loss function in machine learning.

Masagosa [19] showed that the second-order PAC-Bayesian generalization error is especially useful under misspecified models, i.e., for any $\theta$, $p(x|\theta) \neq \nu(x)$. Our theories can also be extended to such a setting (see Appendix H for further discussion).

Other closely related work is an analysis of the weighted majority vote in multiclass classification [20], which uses a second-order PAC-Bayesian bound. While their analysis is specific to the majority vote of multiclass classification, our analysis has been carried out in a more general setting based on the second-order Jensen inequality derived from Theorem 2 and Theorem 3.

# 5 Numerical experiments

According to our Theorem 4, it is important to control the trade-off between the model fitting and diversity enhancement in order to reduce the generalization error. Therefore, we minimize our generalization error bound Eq.(15) directly and confirm that the trade-off is controlled appropriately. Our objective function is

$$\mathcal{F}(\{\theta_i\}_{i=1}^N) := -\frac{1}{N}\sum_{i=1}^N\sum_{d=1}^D [\ln p(x_d|\theta_i) + R(x_d,h)] + \mathrm{KL}(\rho_E, \pi), \tag{25}$$

which is obtained setting $q(\theta) = \rho_E(\theta)$ and $c = 1$ in Eq.(15). We call our approach as Variance regularization, and express it as VAR. We compared the performances of our methods with MAP, $\mathrm{PAC}_E^2$, and f-PVI(f-SVGD) on toy data and real data. Our experimental settings are almost same as that of the previous work [28] and the detailed settings and additional results are shown in Appendix I.

As for the KL divergence, since $\rho_E$ is an empirical measure, we need to carefully define a prior distribution so that the KL divergence between $\rho_E$ and $\pi$ can be defined properly. We used a certain prior distribution as introduced previously [19] (see Appendix C.3 for details). Moreover, to eliminate such a limitation, we considered smoothing the gradient using SVGD. We call this approach

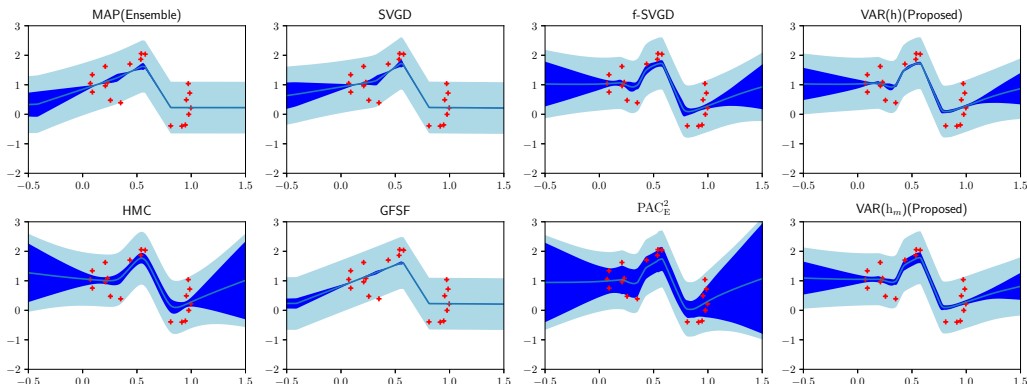

Figure 1: Uncertainty of the regressions. Blue curve is the predictive mean, light shaded area is the predictive cretible interval and dark shaded area is the credible intervals for the mean estimate respectively.

VAR-SVGD, of which update equation is given as

$$\theta_j^{\text{new}} \leftarrow \theta_j^{\text{old}} + \eta \frac{1}{N} \sum_{i=1}^{N} K_{ij} \partial_{\theta_i^{\text{old}}} \left( \log p(\mathcal{D}|\theta_i^{\text{old}}) \pi(\theta_i^{\text{old}}) + R(\mathcal{D}, h) \right) + \partial_{\theta_i^{\text{old}}} K_{ij}, \qquad (26)$$

where $R(\mathcal{D}, h) = \sum_{d=1}^{D} R(x_d, h)$ and $K_{ij}$ is the Gram matrix using the Gaussian kernel defined similarly to the model repulsion in Section 2.1. We tuned its bandwidth by the median trick.

### 5.1 Toy data experiments

First, using toy data, we visualized the model fitting and diversity in various methods. We considered a regression task and we randomly generated the 1-dimensional data points and fit them by using a feed-forward neural network model with 2 hidden layers and ReLU activation, which has 50 units. We used 50 ensembles for each method. The results are shown in Figure 1, which visualizes $95\%$ credible intervals for the prediction and mean estimate corresponding to aleatoric and epistemic uncertainties.

Hamilton Monte Carlo (HMC)[21] is the baseline method used to express the Bayesian predictive uncertainty properly. MAP and SVGD methods give an evaluation of the uncertainty that is too small. Our method and f-SVGD showed very similar evaluations for uncertainty. $\text{PAC}_{\text{E}}^2$ [19] showed large epistemic uncertainty, which is expressed as dark shaded area in Figure 1. We conjecture it is because $\text{PAC}_{\text{E}}^2$ includes the predictive variance in the objective function and the enhanced diversity is too large. Described below, $\text{PAC}_{\text{E}}^2$ shows slightly worse results in real data experiments than those of other approaches. This might be because including the predictive variance in the objective function does not result in a better trade-off of between model fitting and enhancing diversity in practice.

### 5.2 Regression task on UCI

We did regression tasks on the UCI dataset [4]. The model is a single-layer network with ReLU activation and 50 hidden units except for Protein data, which has 100 units. We used 20 ensembles. Results of 20 repetition are shown in Table 2. We found that our method compares favorably with f-SVGD. We also found that $\text{PAC}_{\text{E}}^2$ shows worse performance than those of other methods. We conjectured that this is because the predictive variance in the objective function enhances too large diversity as shown in Figure 1, which hampers the fitting of each model to the data.

### 5.3 Classification task on MNIST and CIFAR 10

We conducted numerical experiments on MNIST and CIFAR 10 datasets. For MNIST, we used a feed-forward network having two hidden layers with 400 units and a ReLU activation function and used 10 ensembles. For CIFAR 10, we used ResNet-32 [9], and we used 6 ensembles. The results are shown

Table 2: Benchmark results on test RMSE and negative log likelihood for the regression task.

| Dataset | Avg. Test RMSE | | | | | Avg. Test negative log likelihood | | | | |
|---|---|---|---|---|---|---|---|---|---|---|
| | MAP | $PAC_E^2$ | f-SVGD | VAR | VAR-SVGD | MAP | $PAC_E^2$ | f-SVGD | VAR | VAR-SVGD |
| Concrete | 5.19±0.3 | 5.49±0.3 | **4.32±0.1** | 4.33±0.1 | 4.35±0.2 | 3.11±0.12 | 3.16±0.10 | 2.86±0.02 | 2.82±0.09 | **2.80±0.06** |
| Boston | 2.98±0.4 | 4.03±0.5 | 2.54±0.3 | 2.54±0.3 | **2.52±0.3** | 2.62±0.2 | 2.61±0.3 | 2.46±0.1 | 2.39±0.2 | **2.35±0.2** |
| Wine | 0.65±0.04 | 1.03±0.09 | **0.61±0.03** | **0.61±0.03** | **0.61±0.03** | 0.97±0.07 | 1.26±0.03 | 0.90±0.05 | **0.89±0.04** | **0.89±0.06** |
| Power | 3.94±0.03 | 5.04±0.21 | 3.77±0.03 | 3.76±0.03 | **3.40±0.05** | 2.79±0.05 | 3.17±0.01 | **2.76±0.05** | 2.79±0.03 | 2.79±0.03 |
| Yacht | 0.86±0.05 | 0.70±0.21 | 0.59±0.09 | 0.59±0.09 | **0.58±0.12** | 1.23±0.05 | **0.80±0.4** | 0.96±0.3 | 0.87±0.3 | 0.81±0.2 |
| Protein | 4.25±0.07 | 4.17±0.05 | 3.98±0.03 | 3.95±0.05 | **3.93±0.07** | 2.95±0.00 | 2.84±0.01 | **2.80±0.01** | 2.81±0.01 | **2.80±0.01** |

Table 3: Benchmark results on test accuracy and negative log likelihood for the classification task.

| Dataset | Test Accuracy | | | | | Test log likelihood | | | | |
|---|---|---|---|---|---|---|---|---|---|---|
| | MAP | $PAC_E^2$ | f-SVGD | VAR | VAR-SVGD | MAP | $PAC_E^2$ | f-SVGD | VAR | VAR-SVGD |
| MNIST | 0.981 | 0.986 | 0.987 | **0.988** | **0.988** | 0.057 | 0.042 | 0.043 | **0.040** | 0.041 |
| CIFAR 10 | **0.935** | 0.919 | 0.927 | 0.928 | 0.928 | **0.215** | 0.270 | 0.241 | 0.238 | 0.240 |

Table 4: Cumulative regret relative to that of the uniform sampling.

| Dataset | MAP | $PAC_E^2$ | f-SVGD | VAR | VAR-SVGD |
|---|---|---|---|---|---|
| Mushroom | 0.129±0.098 | 0.037±0.012 | 0.043±0.009 | **0.029±0.010** | 0.037±0.012 |
| Financial | 0.791±0.219 | 0.189±0.025 | **0.154±0.017** | 0.155±0.024 | 0.176±0.023 |
| Statlog | 0.675 ±0.287 | 0.032±0.0025 | 0.010±0.0003 | **0.006±0.0003** | 0.007±0.0004 |
| CoverType | 0.610±0.051 | 0.396±0.006 | 0.372±0.007 | **0.289±0.003** | 0.320±0.005 |

in Table 3. For both datasets, our methods show competitive performance compared to f-SVGD. For CIFAR 10, as reported previously [28], f-SVGD is worse than the simple ensemble approach. We also evaluated the diversity enhancing property by using the out of distribution performance test. It is hypothesized that Bayesian models are more robust against adversarial examples due to their ability to capture uncertainty. Thus, we generated attack samples and measured the vulnerability to those samples. We found that, as shown in Figure 4 in Appendix I, our method and f-SVGD showed more robustness compared to MAP estimation in each experiment.

### 5.4 Contextual bandit by neural networks on the real data set

Finally, we evaluated the uncertainty of the obtained models using contextual bandit problems [24]. This problem requires the algorithm to balance the trade-off between the exploitation and exploration, and poorly evaluated uncertainty results in larger cumulative regret. We consider the Thompson sampling algorithm with Bayesian neural networks having 2 hidden layers and 100 ReLU units, and we used 20 particles for each experiment. Results of 10 repetition are shown in Table 4. We can see that our approach outperform other methods.

## 6 Conclusion

In this work, we derived a novel second-order Jensen inequality that includes the variance of loss functions. We also derived a PAC-Bayesian generalization error bound. Our error bound shows that both model fitting and enhancing diversity are important for reducing the generalization error. Then, we derived the existing PVI and DPPs from our new Jensen inequality. Finally, we proposed a new PVI that directly minimizes our PAC-Bayesian bound. It shows competitive performance with the current state-of-the-art PVI. In future work, it would be interesting to apply our second-order Jensen inequality to general variational inference or optimal control problems.

Other interesting direction is to derive an upper-bound of the Jensen gap. Gao et al. [6] derived $\ln \mathbb{E}p(x|\theta) - \mathbb{E}\ln p(x|\theta) \leq \text{Variance}(p(x|\theta))$, which uses the predictive distribution. This means that the larger the predictive variance is, the larger upper bound of the Jensen gap we have. We leave it for future work to upper-bound the Jensen gap with the variance of loss function using our theory.

## Acknowledgments and Disclosure of Funding

FF was supported by JST ACT-X Grant Number JPMJAX190R.

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
