## Symbolslist

| Sign | Description |
|------|-------------|
| $\nu(x)$ | Unknown data generating distribution |
| $\mathcal{D} = (x_1, \ldots, x_D)$ | Training data drawn from $\nu(x)$ |
| $p(x|\theta)$ | A model, also called as a likelihood function |
| $\pi(\theta)$ | A prior distribution |
| $p(\theta|\mathcal{D})$ | Bayesian posteriror distribution, $p(\theta|\mathcal{D}) \propto p(\mathcal{D}|\theta)\pi(\theta)$ |
| $q(\theta)$ | An approximate posteriror distribution |
| $\rho_{\mathrm{E}}(\theta)$ | An empirical distribution defined as $\rho_{\mathrm{E}}(\theta) := \frac{1}{N}\sum_{i=1}^{N} \delta_{\theta_i}(\theta)$ |
| $N$ | A number of particles in PVI |
| $\mathcal{F}$ | The objective function of our new PVI |
| $h$ | The bandwidth in the repulsion forces |

## A    Limitation of this work

Here, we discuss the limitation of this work.

Theoretically, the major theoretical limitation is that for all $x$ and $\theta$, $p(x|\theta) < \infty$. This is the exactly same assumption with the previous work [19]. This assumption is always satisfied in classification tasks. However, if $p(x|\theta)$ is a Gaussian distribution, we need to restrict the parameter space such that its variance is larger than $0$. Although there were no problems in our numerical experiments, it cannot be completely denied that in some cases it is possible that the numerical instability of the distribution $p(x|\theta)$ may prevent us from obtaining results consistent with the theory.

## B    Negative societal impacts of this work

Our method uses the ensemble of models, which requires the larger computational cost than the single models.

## C    Further preliminary of existing methods

In this section, we review existing methods.

### C.1    Function space PVI

Here we review the Function space PVI (f-PVI) [28]. Pairs of input-output data are expressed as $\mathcal{D} = \{(x, y)\}$ where $x \in \mathcal{X}$ and $y \in \mathcal{Y}$. We consider the model $p(y|x, \theta) = p(y|f(x; \theta))$ where $f(x; \theta)$ is a $c$-dimensional output function parametrized by $\theta$ and $x$ is an input. For example, for regression tasks, if $\mathcal{Y} = \mathbb{R}$, then $c = 1$ and we often assume that $p(y|x, \theta) = N(y|f(x; \theta), \sigma^2)$. For classification tasks, $c$ corresponds to the class number and $p(y|x, \theta) = \mathrm{Multinomial}(y|\mathrm{softmax}(f(x; \theta)))$.

In previous work [28], they considered that there exists a mapping from the parameters $\theta$ to a function $f(\cdot, \theta)$, and a prior distribution on $\theta$ implicitly defines a prior distribution on the space of the function, $\pi(f)$. Then, the model $p(y|x, \theta)$ corresponds to the distribution of $p(y|x, f)$. Thus, it is possible to obtain the posterior distribution for function $f$ from the inference of the parameters.

Then, we approximate the distribution on $f$ by a size-N ensemble set $\{f_i(\cdot)\}_{i=1}^{N}$. That is, we directly update each $f_i(\cdot)$. Then, the update of f-PVI is given as

$$f_{\mathrm{new}}^{i}(\mathcal{X}) \leftarrow f_{\mathrm{old}}^{i}(\mathcal{X}) + \eta v(\{f_i(\mathcal{X})\}_{i=1}^{N}), \tag{27}$$

where $v$ is the update direction.

However, if the input space $\mathcal{X}$ is very large or infinite, it is impossible to directly update $f_i(\cdot)$ efficiently. Instead, in the previous work [28], we approximate $f_i(\cdot)$ by a parametrized neural network. In principle, it is possible to use any flexible network to approximate it, and it was proposed to use the original network, $f(\cdot; \theta)$ for that approximation since it can express any function on the support of the prior $\pi(f)$ which is implicitly defined by the prior $\pi(\theta)$. Thus, the update direction is mapped to the parameter space

$$\theta_i^{\text{new}} \leftarrow \theta_i^{\text{old}} + \eta \frac{\partial f_i(\mathcal{X})}{\partial \theta_i}\bigg|_{\theta_i = \theta_i^{\text{old}}} v(\{f_i(\mathcal{X})\}_{i=1}^N). \tag{28}$$

Furthermore, it was proposed to replace $\mathcal{X}$ in Eq.(28) by a finite set of samples $\boldsymbol{x}_{1:b} = (x_1, \ldots, x_b)$ with size $b$, which are drawn from $\mathcal{X}^{\otimes b}$. When we input the minibatch with size $b$ into the model, we express it as $f_i(\boldsymbol{x}_{1:b}) = (f_i(x_1), \ldots, f_i(x_b)) \in \mathbb{R}^{cb}$. Then the update equation is given as

$$\theta_i^{\text{new}} \leftarrow \theta_i^{\text{old}} + \eta \frac{\partial f_i(\boldsymbol{x}_{1:b})}{\partial \theta_i}\bigg|_{\theta_i = \theta_i^{\text{old}}} v(\{f_i(\boldsymbol{x}_{1:b})\}_{i=1}^N), \tag{29}$$

where $v(\{f_i\})$ is obtained by replacing $\ln p(\mathcal{D}|\theta)\pi(\theta)$ with $(D/b)\sum_{d=1}^b \ln p(x_d|\theta)\pi(f)$, where $\pi(f)$ is a prior distribution over $f$ and the Gram matrix $K(\theta_i, \theta_j)$ is replaced with $K(f_i^b, f_j^b)$ in Table 1. Thus, f-PVI modifies the loss signal so that models are diverse. We refer to the repulsion term of f-PVI as a model repulsion. When we use the Gaussian kernel, the model repulsion is expressed as

$$\partial_{\theta_i} K(f_i^b, f_j^b) = -h^{-2}(f_i^b - f_j^b)e^{-\|f_i^b - f_j^b\|^2/(2h^2)}\partial_{\theta_i} f_i^b. \tag{30}$$

Thus, the model repulsion pushes model $f_i$ away from $f_j$.

Finally, the implicitly defined prior $\pi(f)$ should be specified more explicitly to calculate $\partial_f \ln \pi(f)$. Then it was proposed approximating the implicit prior by a Gaussian process. Given a minibatch of data $\boldsymbol{x}$, first, we draw parameters from a prior $\pi(\theta)$ and then construct a multivariate Gaussian distribution whose mean and variance are defined by the mean and variance of drawn samples.

## C.2 The second-order PAC-Bayesian generalization error bound [19]

We first present the second-order PAC-Baysian bound in previous work [19],

**Theorem 7.** *[19] For all $x, \theta$, $p(x|\theta) < \infty$ and for any prior distribution $\pi$ over $\Theta$ independent of $\mathcal{D}$ and for any $\xi \in (0, 1)$ and $c > 0$, with probability at least $1 - \xi$ over the choice of training data $\mathcal{D} \sim \nu^{\otimes D}(x)$, for all probability distributions $q$ over $\Theta$, we have*

$$\text{CE} \leq -\mathbb{E}_{\nu,q}[\ln p(x|\theta) + V(x)] \leq \mathbb{E}_q \frac{-1}{D}\sum_{d=1}^D [\ln p(x_d|\theta) + V(x_d)] + \frac{\text{KL}(q, \pi) + \frac{\ln \xi^{-1} + \Psi'_{\pi,\nu}(c,D)}{2}}{cD}, \tag{31}$$

*where*

$$V(x) := (2 \max_\theta p(x|\theta)^2)^{-1}\mathbb{E}_{q(\theta)}\left[(p(x|\theta) - \mathbb{E}_{q(\theta)}p(x|\theta))^2\right], \tag{32}$$

*and*

$$\Psi''_{\pi,\nu}(c, D) := \ln \mathbb{E}_{\pi(\theta,\theta')}\mathbb{E}_{\mathcal{D} \sim \nu^{\otimes D}(x)} e^{cD(-\mathbb{E}_{\nu(x)}L(x,\theta,\theta') + D^{-1}\sum_{d=1}^D L(x_d,\theta,\theta'))}, \tag{33}$$

*and*

$$L(x, \theta, \theta') := \ln p(x|\theta) + (2 \max_\theta p(x|\theta)^2)^{-1}(p(x|\theta)^2 - p(x|\theta)p(x|\theta')). \tag{34}$$

## C.3 Ensemble setting of the second-order PAC-Bayesian generalization error bound [19]

Here we introduce the ensemble setting, that is,

$$\rho_{\text{E}}(\theta) = \frac{1}{N}\sum_{i=1}^N \delta_{\theta_i}(\theta). \tag{35}$$

### C.3.1 Prior distribution for ensemble learning

We need to properly define a prior distribution for ensemble learning so that KL divergence between $\rho_E$ and $\pi$ can be defined properly. Following the previous work [19], we define the prior distribution as the mixture of discrete dirac mass distribution as

$$\pi_E(\theta) = \sum_{\theta' \in \Theta_E} w_{\theta'} \delta_{\theta'}(\theta), \tag{36}$$

where $w_{\theta'} \geq 0$ and $\sum_{\theta' \in \Theta_E} w_{\theta'} = 1$.

If

$$\{\theta_i\}_{i=1}^N \subset \Theta_E \subset \mathbb{R}^d, \tag{37}$$

holds, we can define the KL divergence properly. From the definition of Radon–Nikodym derivative of the discrete measure, we have

$$\mathrm{KL}(\rho_E, \pi_E) = \frac{1}{N} \sum_{i=1}^N \ln \frac{\frac{1}{N}}{w_{\theta_i}} = -\frac{1}{N} \sum_{i=1}^N \ln \pi_E(\theta_i) + \frac{1}{N} \sum_{i=1}^N \ln \frac{1}{N}. \tag{38}$$

Thus, we need to properly define $\Theta_E$ so that Eq.(35) holds.

Following the idea of [19], we define $\Theta_E$ as the set of $d$-dimensional real vectors that can be represented under a finite-precision scheme using $p$-bits to encode each element of the vector. Thus this set is countable and can define the mixture of dirac mass distribution and satisfies Eq.(35) properly.

Note that as discussed in the previous work [19], the KL divergence of Eq.(38) is not continuous and differentiable and not suitable for the gradient descent based optimization. Fortunately, when we implement any statistical distribution on a computer, they are expressed under a finite-precision scheme, thus, we can regard them as an approximation of $\pi_E$. Thus, we can use any statistical distribution as a proxy of a precise $\pi_E$ when we implement algorithms on a computer.

### C.3.2 Generalization error bound

Using the prior distribution $\pi_E$ introduced in Appendix C.3.1, we have the PAC-Bayesian generalization error bound for the ensemble setting,

**Theorem 8.** *[19] For all $x, \theta$, $p(x|\theta) < \infty$ and for any prior distribution $\pi_E$ over $\Theta_E$ independent of $\mathcal{D}$ and for any $\xi \in (0,1)$ and $c > 0$, with probability at least $1 - \xi$ over the choice of training data $\mathcal{D} \sim \nu^{\otimes D}(x)$, for all probability distributions $\rho_E$ with $\mathrm{supp}(\rho_E) \subset \Theta_E$, we have*

$$\mathrm{CE} \leq -\mathbb{E}_{\nu,\rho_E}[\ln p(x|\theta) + V(x)] \leq \mathbb{E}_{\rho_E} \frac{-1}{D} \sum_{d=1}^D [\ln p(x_d|\theta) + V(x_d)] + \frac{\mathrm{KL}(\rho_E, \pi_E) + \frac{\ln \xi^{-1} + \Psi'_{\pi_E,\nu}(c,D)}{2}}{cD}, \tag{39}$$

*where*

$$V(x) := (2 \max_\theta p(x|\theta)^2)^{-1} \mathbb{E}_{\rho_E} \left[ (p(x|\theta) - \mathbb{E}_{\rho_E} p(x|\theta))^2 \right], \tag{40}$$

*and*

$$\Psi''_{\pi_E,\nu}(c,D) := \ln \mathbb{E}_{\pi(\theta,\theta')} \mathbb{E}_{\mathcal{D} \sim \nu^{\otimes D}(x)} \mathrm{e}^{cD(-\mathbb{E}_{\nu(x)} L(x,\theta,\theta') + D^{-1} \sum_{d=1}^D L(x_d,\theta,\theta'))}, \tag{41}$$

*and*

$$L(x, \theta, \theta') := \ln p(x|\theta) + (2 \max_\theta p(x|\theta)^2)^{-1} (p(x|\theta)^2 - p(x|\theta)p(x|\theta')). \tag{42}$$

## D  Proofs of Section 3

### D.1  Proof of Theorem 2

Since $\psi$ is a monotonically increasing concave function on $\mathbb{R}$, an inverse function $\psi^{-1}$ is convex function.

Let us consider the Taylor expansion of $\psi$ up to the second order around a constant $\mu$. There exists a constant $c(z)$ between $\mu$ and $z$ that satisfies

$$\psi(z) = \psi(\mu) + \psi'(\mu)(z - \mu) + \frac{\psi''(c(z))}{2}(z - \mu)^2. \tag{43}$$

Above equation holds for all realizable values of a random variable $Z$.

Then, we substitute a random variable $Z$ to $z$ and $\mu = \psi^{-1}(\mathbb{E}_{p_Z}[\psi(Z)])$ in the above equation, and then taking the expectation with respect to $p_Z(z)$, we obtain

$$0 = \psi'(\mu) \int_{\mathbb{R}^+} (z - \mu)p_Z(z)dz + \int_{\mathbb{R}^+} \frac{\psi''(c(z))}{2}(z - \mu)^2 p_Z(z)dz. \tag{44}$$

By rearranging the above, and $\psi' > 0$ since $\psi'$ is a monotonically increasing function, we have

$$\mathbb{E}_{p_Z} Z = \mu - \int_{\mathbb{R}^+} \frac{\psi''(c(z))}{2\psi'(\mu)}(z - \mu)^2 p_Z(z)dz \tag{45}$$

Since $\mathbb{E}_{p_Z} Z \in \mathbb{R}^+$ from the assumption and since $\psi$ is a monotonically increasing function, $\psi'$ is always positive and $\psi''$ is always negative. Thus $\mu - \mathbb{E}_{p_Z} \frac{\psi''(c)}{2\psi'(\mu)}(Z - \mu)^2 \in \mathbb{R}^+$. Then we apply the $\psi$ on both hand side, we obtain the theorem.

### D.2   Proof of Corollary 1

**Remark 7.** *This corollary holds for all probability distributions $q$ over $\Theta$, like Theorem 1.*

*Proof.* We substitute $\psi = \log$ and $Z = p(x|\theta)$ in Theorem 2, we obtain the result. Here we also show the more intuitive proof.

Let us consider the Taylor expansion of log function up to the second order around a constant $\mu = \mathbb{E}_q \ln p(x|\theta)$. We obtain

$$\ln e^{\ln p(x|\theta)} = \ln e^{\mathbb{E}_q \ln p(x|\theta)} + \frac{1}{e^{\mathbb{E}_q \ln p(x|\theta)}}(e^{\ln p(x|\theta)} - e^{\mathbb{E}_q \ln p(x|\theta)}) - \frac{1}{2g(x,\theta)^2}(e^{\ln p(x|\theta)} - e^{\mathbb{E}_q \ln p(x|\theta)})^2 \tag{46}$$

where $g(x, \theta)$ is the constant between $p(x|\theta)$ and $\mu$, and it is defined as the reminder of the Taylor expansion. Taking the expectation, we have

$$0 = \mathbb{E}_q \frac{1}{e^{\mathbb{E}_q \ln p(x|\theta)}}(e^{\ln p(x|\theta)} - e^{\mathbb{E}_q \ln p(x|\theta)}) - \mathbb{E}_q \frac{1}{2g(x,\theta)^2}(e^{\ln p(x|\theta)} - e^{\mathbb{E}_q \ln p(x|\theta)})^2 \tag{47}$$

We rearrange the equality as follows

$$e^{\mathbb{E}_q \ln p(x|\theta)} \left( 1 + \mathbb{E}_q \frac{1}{2g(x,\theta)^2}(e^{\ln p(x|\theta)} - e^{\mathbb{E}_q \ln p(x|\theta)})^2 \right) = \mathbb{E}_q e^{\ln p(x|\theta)}. \tag{48}$$

Then taking the logarithm in both hand side, we obtain the result. □

### D.3   Proof of Theorem 3

**Remark 8.** *This theorem holds for all probability distributions $q$ over $\Theta$, like Theorem 1.*

*Proof.* First, recall that $g(x, \theta)$ is the constant between $p(x|\theta)$ and $\mu$ is defined as the reminder of the second order Taylor expansion. Thus if we define $g(x) := \max_{\theta \in \text{supp}(q(\theta))} p(x|\theta)$, then $g(x) \geq g(x, \theta)$ holds. Moreover, following relation holds:

$$\frac{1}{2} \left( \frac{e^{\ln p(x|\theta)} - e^{\mathbb{E}_q \ln p(x|\theta)}}{g(x)} \right)^2 \leq 1. \tag{49}$$

Using the following lemma (its proof is shown in the below)

**Lemma 1.** *For any constant $\alpha \in (0, 1]$, we have*

$$-\ln(1 + \alpha) \leq \ln(1 - \frac{\alpha}{2}). \tag{50}$$

Then we obtain

$$\mathbb{E}_q \ln p(x|\theta) \leq \ln \mathbb{E}_q p(x|\theta) + \ln\left(1 - \mathbb{E}_q \frac{1}{4g(x)^2}(e^{\ln p(x|\theta)} - e^{\mathbb{E}_q \ln p(x|\theta)})^2\right). \tag{51}$$

Then using the relation

$$1 - \frac{\alpha}{2} \leq e^{-\frac{\alpha}{2}}, \tag{52}$$

we get

$$\mathbb{E}_q \ln p(x|\theta) \leq \ln \mathbb{E}_q p(x|\theta) + \ln e^{-\mathbb{E}_q\left(\frac{e^{\ln p(x|\theta)} - e^{\mathbb{E}_q \ln p(x|\theta)}}{2g}\right)^2}. \tag{53}$$

Finally, we use the following lemma (the proof is shown in the below),

**Lemma 2.** *For any positive constants $\alpha, \beta > 0$, we have*

$$\sqrt{\alpha\beta} \leq \frac{\alpha - \beta}{\ln \alpha - \ln \beta}, \tag{54}$$

*and when $\alpha = \beta$, the equality holds.*

Above lemma is equivalent to

$$(\ln \alpha - \ln \beta)^2 \alpha\beta \leq (\alpha - \beta)^2. \tag{55}$$

Setting $\alpha := e^{\ln p(x|\theta)}$ and $\beta := e^{\mathbb{E}_q \ln p(x|\theta)}$ and substituting them into Eq.(55), we obtain

$$(\ln p(x|\theta) - \mathbb{E}_q \ln p(x|\theta))^2 p(x|\theta)e^{\mathbb{E}_q \ln p(x|\theta)} \leq (e^{\ln p(x|\theta)} - e^{\mathbb{E}_q \ln p(x|\theta)})^2. \tag{56}$$

Then we obtain

$$\mathbb{E}_q \ln p(x|\theta) \leq \ln \mathbb{E}_q p(x|\theta) + \ln e^{-\mathbb{E}_q p(x|\theta)e^{\mathbb{E}_q \ln p(x|\theta)}\left(\frac{(\ln p(x|\theta) - \mathbb{E}_q \ln p(x|\theta))}{2g(x)}\right)^2}. \tag{57}$$

We define the bandwidth as

$$p(x|\theta)e^{\mathbb{E}_q \ln p(x|\theta)}/g(x)^2 := h(x, \theta)^{-2}, \tag{58}$$

then, we have

$$\mathbb{E}_q \ln p(x|\theta) \leq \ln \mathbb{E}_q p(x|\theta) + \ln e^{-\mathbb{E}_q\left(\frac{\ln p(x|\theta) - \mathbb{E}_q \ln p(x|\theta)}{2h(x, \theta)}\right)^2}. \tag{59}$$

□

### D.3.1 Proof of lemma 1

*Proof.* Define $f(\alpha) := \ln(1 - \frac{\alpha}{2}) + \ln(1 + \alpha)$. Since $f'(\alpha) > 0$ for $0 \leq \alpha < 1$, thus $0 = f(0) \leq f(\alpha)$. This concludes the proof. □

### D.3.2 Proof of lemma 2

*Proof.* Since

$$\frac{\alpha - \beta}{\ln \alpha - \ln \beta} \geq 0, \tag{60}$$

we only need to show

$$\frac{\ln \alpha - \ln \beta}{\alpha - \beta} \leq \frac{1}{\sqrt{\alpha\beta}}. \tag{61}$$

Since this inequality is symmetric with respect to $\alpha$ and $\beta$, we can assume that $\alpha \geq \beta$. If $\alpha = \beta$, it is clear by setting $\beta = \alpha + \epsilon$ where $\epsilon > 0$ and take the limit to $\epsilon \to 0^+$. When $\alpha > \beta$, we define $\alpha = t^2\beta$ where $t^2 > 1$ and $t > 1$. Substituting this assumption in the above, we need to show that

$$\frac{2\ln t}{t^2 - 1} \leq \frac{1}{t}. \tag{62}$$

Since $t^2 - 1 > 0$, by rearranging the above inequality, we only need to show that

$$t - \frac{1}{t} - 2\ln t \geq 0. \tag{63}$$

Thus, we define $f(t) := t^2 - 1 - 2t\ln t$. Then $f'(t) = 2(t - 1 - \ln t) \geq 0$ for all $t$, thus $0 = f(1) \leq f(t)$ for all $t \geq 1$. Then, $t - \frac{1}{t} - 2\ln t = \frac{1}{t}f(t)$, we have shown Eq.(63). This concludes the proof. $\qquad\square$

### D.4 Proof of Theorem 4

We first present the complete statement:

**Theorem 9.** *For all $x$ and $\theta$, $p(x|\theta) < \infty$ and for any prior distribution $\pi$ over $\Theta$ independent of $\mathcal{D}$ and for any $\xi \in (0,1)$ and $c > 0$, with probability at least $1 - \xi$ over the choice of training data $\mathcal{D} \sim \nu^{\otimes D}(x)$, for all probability distribution $q$ over $\Theta$, we have*

$$\mathrm{CE} \leq -\mathbb{E}_{\nu,q}[\ln p(x|\theta) + R(x,h)] \leq \mathbb{E}_q \frac{-1}{D} \sum_{d=1}^{D} [\ln p(x_d|\theta) + R(x_d,h_m)] + \frac{\mathrm{KL}(q,\pi) + \frac{\ln \xi^{-1} + \Psi''_{\pi,\nu}(c,D)}{3}}{cD}, \tag{64}$$

*where*

$$\Psi''_{\pi,\nu}(c,D) := \ln \mathbb{E}_{\pi(\theta,\theta',\theta'')} \mathbb{E}_{\mathcal{D}\sim\nu^{\otimes D}(x)} e^{cD(-\mathbb{E}_{\nu(x)}L(x,\theta,\theta',\theta'')+D^{-1}\sum_{d=1}^{D}L(x_d,\theta,\theta',\theta''))}, \tag{65}$$

*and*

$$L(x,\theta,\theta',\theta'') := \ln p(x|\theta) + (2h_m(x,\theta))^{-2}((\ln p(x|\theta))^2 - 2\ln p(x|\theta)\ln p(x|\theta') + \ln p(x|\theta')\ln p(x|\theta'')). \tag{66}$$

*Proof.* From the definition of the band width, we have

$$\mathrm{CE} \leq -\mathbb{E}_{\nu,q}[\ln p(x|\theta) + R(x,h)] \leq -\mathbb{E}_{\nu,q}[\ln p(x|\theta) + R(x,h_m)]. \tag{67}$$

Thus, our goal is to derive the probabilistic relationship

$$-\mathbb{E}_{\nu,q}[\ln p(x|\theta) + R(x,h_m)] \approx \mathbb{E}_q \frac{-1}{D} \sum_{d=1}^{D} [\ln p(x_d|\theta) + R(x_d,h_m)]. \tag{68}$$

We express $\mathbb{E}_{q(\theta)}[\ln p(x|\theta) + R(x,h_m)]$ as

$$\mathbb{E}_{q(\theta)q(\theta')q(\theta'')}L(x,\theta,\theta',\theta''), \tag{69}$$

where

$$L(x,\theta,\theta',\theta'') := \ln p(x|\theta) + (2h_m(x,\theta))^{-2}((\ln p(x|\theta))^2 - 2\ln p(x|\theta)\ln p(x|\theta') + \ln p(x|\theta')\ln p(x|\theta'')). \tag{70}$$

Then, we consider the same proof as Pac-Bayesian bound of Theorem 1 [7] to this loss function. Applying Theorem 3 in [7] with a prior $\pi(\theta,\theta',\theta'') := \pi(\theta)\pi(\theta')\pi(\theta'')$, we have

$$\mathbb{E}_{\nu(x),q(\theta)q(\theta')q(\theta'')}L(x,\theta,\theta',(\theta'') \leq -\frac{1}{D}\sum_{d}^{D}\mathbb{E}_{q(\theta)q(\theta')q(\theta'')}L(x_d,\theta,\theta',\theta'')$$

$$+ \frac{\mathrm{KL}(q(\theta)q(\theta')q(\theta'')|\pi(\theta,\theta',\theta'')) + \ln\xi^{-1} + \Psi''_{\pi,\nu}(\lambda,D)}{\lambda}, \tag{71}$$

where

$$\Psi''_{\pi,\nu}(c,D) := \ln \mathbb{E}_{\pi(\theta,\theta',\theta'')} \mathbb{E}_{\mathcal{D}\sim\nu^{\otimes D}(x)} e^{cD(-\mathbb{E}_{\nu(x)}L(x,\theta,\theta',\theta'')+D^{-1}\sum_{d=1}^{D}L(x_d,\theta,\theta',\theta''))}. \tag{72}$$

Noting that $\mathrm{KL}(q(\theta)q(\theta')q(\theta'')|\pi(\theta,\theta',\theta'')) = 3\mathrm{KL}(q(\theta)|\pi(\theta))$, reparametrizing $\lambda = 3cD$, we obtain the main result. $\qquad\square$

### D.4.1 Median lower bound

The goal of this section is to show that if $\mathbb{E}_q[\ln p(x|\theta)]^2 < M < \infty$, then we have

$$\text{Med}[e^{\ln p(x|\theta)}]e^{-M^{1/2}} \leq e^{\mathbb{E}_q \ln p(x|\theta)}, \tag{73}$$

where Med is the median of the random variable. We relax the condition of the bandwidth $h_m$ in Theorem 4. To derive Theorem 4, we lower-bound $e^{\mathbb{E}_q \ln p(x|\theta)}$ by $e^{\min_\theta p(x|\theta)}$ and introduced $h_m$. This $e^{\min_\theta p(x|\theta)}$ can be a small value for many practical models. If we can use Eq.(73), we can replace $e^{\min_\theta p(x|\theta)}$ in $h_m$ with $\text{Med}[e^{\ln p(x|\theta)}]e^{-M^{1/2}}$, which results in a much tighter bound.

*Proof.* We use the following lemma in the previous work [11],

**Lemma 3.** *Given a random variable $\mathbb{E}X^2 < \infty$, we have*

$$|\mathbb{E}[X] - \text{Med}[X]| \leq \sqrt{\mathbb{E}(X - \mathbb{E}X)^2}. \tag{74}$$

For the completeness, we show its proof.

*Proof.* Since the median minimizes the mean absolute value error, we have

$$|\mathbb{E}[X] - \text{Med}[X]| = |\mathbb{E}[X - \text{Med}[X]]| \leq \mathbb{E}|X - \text{Med}[X]| \leq \mathbb{E}|X - \mathbb{E}X| \leq \sqrt{\mathbb{E}(X - \mathbb{E}X)^2}. \tag{75}$$

$\square$

From this lemma, we have

$$\text{Med}[X] - \sqrt{\mathbb{E}(X - \mathbb{E}X)^2} \leq \mathbb{E}X, \tag{76}$$

and taking the exponential, we have

$$e^{\text{Med}[X]}e^{-\sqrt{\mathbb{E}(X-\mathbb{E}X)^2}} \leq e^{\mathbb{E}X}. \tag{77}$$

From the definition of the median and the monotonically increasing property of the exponential function, we have

$$e^{\text{Med}[X]} = \text{Med}[e^X]. \tag{78}$$

Thus, we have

$$\text{Med}[e^X]e^{-\sqrt{\mathbb{E}(X-\mathbb{E}X)^2}} \leq e^{\mathbb{E}X}. \tag{79}$$

Finally, by setting $X = \ln p(x|\theta)$, and assume that $\mathbb{E}_q[\ln p(x|\theta)]^2 < M < \infty$, we have

$$\text{Med}[e^{\ln p(x|\theta)}]e^{-\sqrt{\mathbb{E}_q(\ln p(x|\theta) - \mathbb{E}_q \ln p(x|\theta))^2}} \leq e^{\mathbb{E}_q \ln p(x|\theta)}, \tag{80}$$

thus, we have

$$\text{Med}[e^{\ln p(x|\theta)}]e^{-M^{1/2}} \leq e^{\mathbb{E}_q \ln p(x|\theta)}. \tag{81}$$

$\square$

### D.5 Proof of Theorem 5

**Remark 9.** *This theorem and the results of Section 3.2 holds for all probability distributions $\rho_\text{E}$ over $\Theta$ that is expressed as the mixture of the dirac distribution, that is, $\rho_\text{E}(\theta) = \frac{1}{N} \sum_{i=1}^{N} \delta_{\theta_i}(\theta)$.*

*Proof.* First note that, from Theorem 3, by substituting the definition of $\rho_\text{E}$, we have

$$\mathbb{E}_{\rho_\text{E}} \ln p(x|\theta) \leq \ln \mathbb{E}_{\rho_\text{E}} p(x|\theta) - \frac{1}{N} \sum_{i=1}^{N} \left( \frac{\ln p(x|\theta_i) - \frac{1}{N} \sum_{j=1}^{N} \ln p(x|\theta_j)}{2h(x,\theta)} \right)^2. \tag{82}$$

Here the bandwidth $h(x,\theta)$ is

$$h(x,\theta)^{-2} = \frac{e^{\ln p(x|\theta_i) + \frac{1}{N}\sum_{i=1}\ln p(x|\theta_i)}}{e^{2\max_{j\in\{1,\ldots,N\}}\ln p(x|\theta_j)}}.$$  (83)

and we simply express the $\max_{j\in\{1,\ldots,N\}}\ln p(x|\theta_j)$ as $\max_j \ln p(x|\theta_j)$. This is because that the bandwidth $h(x,\theta)$ is derived by the relation $g(x) \geq g(x,\theta)$ in Appendix D.3 and $g(x,\theta)$ is the constant between $p(x|\theta)$ and $e^{\mathbb{E}_q \ln p(x|\theta)}$. Thus, we can upper bound $g(x,\theta)$ by $\max_j \ln p(x|\theta_j)$ since $\rho_{\mathrm{E}}$ takes values only on $\theta_1,\ldots,\theta_N$.

For that purpose, we first eliminate the dependence of $\theta$ from the bandwidth. From the definition, Let us define

$$h_w(x,\theta)^{-2} := \frac{e^{\min_i \ln p(x|\theta_i) + \frac{1}{N}\sum_{i=1}\ln p(x|\theta_i)}}{e^{2\max_j \ln p(x|\theta_j)}},$$  (84)

then we have

$$h_w(x,\theta)^{-2} \leq h(x,\theta)^{-2}.$$  (85)

Thus, we have

$$\mathbb{E}_{\rho_{\mathrm{E}}} \ln p(x|\theta) \leq \ln \mathbb{E}_{\rho_{\mathrm{E}}} p(x|\theta) - \frac{1}{4h_w(x,\theta)^2 N}\sum_{i=1}^{N}\left(\ln p(x|\theta_i) - \frac{1}{N}\sum_{j=1}^{N}\ln p(x|\theta_j)\right)^2.$$  (86)

Next we focus on the following relation for the variance. For simplicity, we express $L_i := \ln p(x|\theta_i)$. Then by rearranging the definition,

$$\frac{1}{N}\sum_{i=1}^{N}(L_i - \sum_{j=1}^{N}\frac{1}{N}L_j)^2$$

$$= \frac{1}{N^3}\sum_{i=1}^{N}\left(N^2 L_i^2 - 2NL_i(\sum_{j=1}^{N}L_j) + (\sum_{j=1}^{N}L_j)^2\right)$$

$$= \frac{1}{N^3}\sum_{i=1}^{N}\left(N^2 L_i^2 - 2NL_i(\sum_{j=1}^{N}L_j) + \sum_{j=1}^{N}L_j^2 + 2\sum_{j=1}^{N-1}L_j L_{j+1}\right)$$

$$= \frac{1}{N^3}\sum_{i=1}^{N}\left(\sum_{j=1}^{N}(L_j - L_i)^2 + (N^2 - N)L_i^2 - 2(N-1)L_i\sum_{j=1}^{N}L_j + 2\sum_{j=1}^{N-1}L_j L_{j+1}\right)$$

$$= \frac{1}{N^3}\left(\sum_{i,j=1}^{N}(L_j - L_i)^2 + (N^2 - N)\sum_{i=1}^{N}L_i^2 - 2(N-1)(\sum_{i=1}^{N}L_i)^2 + 2N\sum_{i=1}^{N-1}L_i L_{i+1}\right)$$

$$= \frac{1}{N^3}\left(\sum_{i,j=1}^{N}(L_j - L_i)^2 + (N^2 - 3N + 2)\sum_{i=1}^{N}L_i^2 + (-2N+4)\sum_{i=1}^{N-1}L_i L_{i+1}\right)$$

$$= \frac{1}{N^3}\left(\sum_{i,j=1}^{N}(L_j - L_i)^2 + (N^2 - 2N)\sum_{i=1}^{N}L_i^2 + (-N+2)\left(\sum_{i=1}^{N}L_i\right)^2\right)$$

$$= \frac{1}{N^3}\sum_{i,j=1}^{N}(L_j - L_i)^2 + \frac{(N-2)}{N}\left(\frac{1}{N}\sum_{i=1}^{N}(L_i)^2 - \left(\sum_{i=1}^{N}\frac{L_i}{N}\right)^2\right)$$

$$= \frac{1}{N^3}\sum_{i,j=1}^{N}(L_j - L_i)^2 + \frac{(N-2)}{N}\left(\frac{1}{N}\sum_{i=1}^{N}(L_i - \sum_{j=1}^{N}\frac{1}{N}L_j)^2\right).$$  (87)

Thus, we have

$$\frac{2}{N}\frac{1}{N}\sum_{i=1}^{N}(L_i - \sum_{j=1}^{N}\frac{1}{N}L_j)^2 = \frac{1}{N^3}\sum_{i,j=1}^{N}(L_j - L_i)^2, \tag{88}$$

which means

$$\frac{1}{N}\sum_{i=1}^{N}(L_i - \sum_{j=1}^{N}\frac{1}{N}L_j)^2 = \frac{1}{2N^2}\sum_{i,j=1}^{N}(L_j - L_i)^2. \tag{89}$$

This concludes the proof. □

### D.6 Proof of Theorem 6

*Proof.* Inspired by the rescaling in the main paper, we use the rescaling $\tilde{h}\ln N$ and define the $N \times N$ kernel matrix $K$ of which the $(i,j)$ element is defined as

$$K_{ij} := \exp\left(-\tilde{h}\ln N(8h_w^2)^{-1}(\ln p(x|\theta_i) - \ln p(x|\theta_j))^2\right). \tag{90}$$

Here, we introduced the additional bandwidth $\tilde{h}\ln N$ to rescale the kernel. As we will see in the proof below, any positive $\tilde{h}$ can be used for the bandwidth. However, using large $\tilde{h}$ makes the bound loose. For simplicity, define $-R_1 := \frac{1}{N^2\tilde{h}\ln N}\sum_{i,j=1}^{N}\ln K_{ij}$.

By applying the Jensen inequality, we have

$$-R_1 = \frac{1}{N^2\tilde{h}\ln N}\sum_{i,j=1}^{N}\ln K_{ij} \leq \frac{1}{\tilde{h}\ln N}\ln\sum_{i,j=1}^{N}\frac{K_{ij}}{N^2}. \tag{91}$$

Then note that from the definition of $K$, we have

$$\sum_{i,j=1}^{N}\frac{K_{ij}}{N^2} \leq 1. \tag{92}$$

We also have

$$-\tilde{h}R_1 \leq \frac{1}{\ln N}\ln\sum_{i,j=1}^{N}\frac{K_{ij}}{N^2} \leq \frac{1}{N}\ln\sum_{i,j=1}^{N}\frac{K_{ij}}{N^2} = -\frac{1}{N}\ln N^2 + \frac{1}{N}\ln\sum_{i,j=1}^{N}K_{ij}. \tag{93}$$

Then we define the new kernel function as

$$\tilde{K}_{ij} = K_{ij}^{1/2}. \tag{94}$$

We use the relation of the Frobenius norm and the trace,

$$\sum_{i,j=1}^{N}K_{ij} = \sum_{i,j}\tilde{K}_{i,j}^2 = \mathrm{Tr}(\tilde{K}^\top\tilde{K}), \tag{95}$$

thus we have

$$\begin{aligned}
-\tilde{h}R_1 &\leq -\frac{1}{N}\ln N^2 + \frac{1}{N}\ln\mathrm{Tr}[\tilde{K}\tilde{K}^\top] \\
&\leq -\frac{1}{N}\ln N^2 + \frac{1}{N}\ln(\mathrm{Tr}[\tilde{K}\tilde{K}^\top] + \epsilon) \\
&\leq -\frac{1}{N}\ln N^2 - \frac{N-1}{N}\ln\epsilon + \frac{1}{N}\ln(\det[\epsilon I + \tilde{K}\tilde{K}^\top]) \\
&\leq -\frac{2}{N}\ln N - \frac{N-1}{N}\ln\epsilon + \frac{2}{N}\ln(\det[\epsilon^{1/2}I + \tilde{K}]).
\end{aligned} \tag{96}$$

From the second line to the third line, we used the following relation. We express the eigenvalues of $\tilde{K}\tilde{K}^\top$ as $\rho_i$s. Then since $\rho_i \geq 0$, we have

$$\det(\epsilon I + \tilde{K}\tilde{K}^\top)$$
$$= \prod_i (\epsilon + \rho_i)$$
$$\geq \epsilon^N + \prod_i \rho_i + \epsilon^{N-1} \sum_i \rho_i$$
$$\geq \epsilon^N + \det[\tilde{K}\tilde{K}^\top] + \epsilon^{N-1}\mathrm{Tr}[\tilde{K}\tilde{K}^\top] \geq \epsilon^{N-1}\mathrm{Tr}[\tilde{K}\tilde{K}^\top], \tag{97}$$

and thus we have

$$\ln \mathrm{Tr}[\tilde{K}\tilde{K}^\top] \leq \ln \det(\epsilon I + \tilde{K}\tilde{K}^\top) - (N-1)\ln \epsilon. \tag{98}$$

Then apply this to the second line in Eq.(103). In the last inequality in Eq.(103), we used the relation

$$\ln(\det[\epsilon^{1/2}I + \tilde{K}]) = \frac{1}{2}\ln(\det[(\epsilon^{1/2}I + \tilde{K})(\epsilon^{1/2}I + \tilde{K})^\top]$$
$$= \frac{1}{2}\ln(\det[(\epsilon I + \epsilon^{1/2}(\tilde{K} + \tilde{K}^\top) + \tilde{K}\tilde{K}^\top]$$
$$\geq \frac{1}{2}\ln(\det[(\epsilon I + \tilde{K}\tilde{K}^\top)]). \tag{99}$$

Thus, we have

$$-\tilde{h}R_1 \leq -\frac{2}{N}\ln N - \frac{N-1}{N}\ln \epsilon + \frac{2}{N}\ln(\det[\epsilon^{1/2}I + \tilde{K}])$$
$$\leq -\frac{2}{N}\ln N - \frac{N-1}{N}\ln \epsilon + \frac{2}{N}\sum_i^N \ln(\epsilon^{1/2} + \lambda_i)$$
$$\leq -\frac{N-1}{N}\ln \epsilon + \frac{2}{N}\sum_i^N \ln \frac{(\epsilon^{1/2} + \lambda_i)}{N}, \tag{100}$$

where $\lambda_i$ is the $i$-th eigenvalue of $\tilde{K}$.

Note that from the definition of the gram matrix, for all $i$, $\tilde{K}_{ii} = K_{ii}^{1/2} = 1$ holds and this means that $\mathrm{Tr}\tilde{K} = \sum_i \rho_i = N$. From the relation of $\ln x \leq x - 1$ and $\sum_i \rho_i = N$, we obtain

$$-\tilde{h}R_1 \leq -\frac{N-1}{N}\ln \epsilon + \frac{2}{N}\sum_i^N \ln \frac{(\epsilon^{1/2} + \lambda_i)}{N}$$
$$\leq -\frac{N-1}{N}\ln \epsilon + \frac{2}{N}\sum_i^N \left(\frac{(\epsilon^{1/2} + \lambda_i)}{N} - 1\right)$$
$$\leq -\frac{N-1}{N}\ln \epsilon + \frac{2}{N}\left(\epsilon^{1/2} + 1 - N\right). \tag{101}$$

Thus, if we set $1 \leq \epsilon \leq (N-1)^2$, the right-hand side of the above is smaller than 0. In conclusion, we have

$$-\tilde{h}R_1 \leq 0 \tag{102}$$

if we set $1 \leq \epsilon \leq (N-1)^2$. Finally, we set $\epsilon^{1/2}$ as $\epsilon$ to make the notation the same as in the theorem statement. This means that for any $1 \leq \epsilon \leq (N-1)$, we have

$$-R_1 \leq -\frac{2}{\tilde{h}N}\ln N + \frac{2}{\tilde{h}N}\ln(\det[\epsilon^{1/2}I + K]), \tag{103}$$

where

$$K_{ij} = \exp\left(-\tilde{h}\ln N(4h_w)^{-2}(\ln p(x|\theta_i) - \ln p(x|\theta_j))^2\right). \tag{104}$$

Next, we discuss the effect of $\tilde{h}$. $\tilde{h}$ influences the eigenvalues $\rho_i$ in the following way. By using the Gershgorin circle theorem [8] and the definition of $\tilde{K}$ for each $i$,

$$\lambda_i \leq \sum_j K_{ij}. \tag{105}$$

Each element $K_{ij}$ depends on $\tilde{h}$, so by choosing the $\tilde{h}$ appropriately, we can make $R_1$ small and the bound becomes tighter.

$\square$

### D.7  Proof of Eq.(23)

Our gram matrix $K$ is symmetric stationary kernel function, and thus it satisfies

$$\partial_{\theta_i} K_{i,j} = -\partial_{\theta_j} K_{i,j}. \tag{106}$$

Then from the log determinant property, we have

$$\partial_{\theta_i} \ln \det(\epsilon I + K) = \mathrm{Tr}\left[(\epsilon I + K)^{-1}\partial_\theta K\right]. \tag{107}$$

Then we have

$$\partial_{\theta_i} \ln \det(\epsilon I + K) = \mathrm{Tr}\left[(\epsilon I + K)^{-1}\partial_\theta K\right] = \sum_{i=1}^{N}(\epsilon I + K)^{-1}_{ij}\partial_{\theta_i} K_{ij} = -\sum_{i=1}^{N}(\epsilon I + K)^{-1}_{ij}\partial_{\theta_j} K_{ij}, \tag{108}$$

and in the second equality, we used the definition of the trace. Then we get

$$\partial_{\theta_i} Obj(\{\theta_i\}) = \frac{1}{N}\partial_{\theta_i} \log p(x|\theta_i) + \frac{2}{\tilde{h}N}\sum_j (K + \epsilon I)^{-1}_{ij}\nabla_{\theta_i} K_{ij}. \tag{109}$$

Note that the small positive constant $c$ of GFSF shown in Table 1 is introduced so that $K + cI$ can have a inverse matrix.

### D.8  Proof of the repulsion to DPP

We define the new kernel function as

$$\tilde{G}_{ij} = G_{ij}^{1/2}, \tag{110}$$

where $G$ is defined in Eq.(17).

We use the relation of the Frobenius norm and the trace,

$$\sum_{i,j=1}^{N} G_{ij} = \sum_{i,j} \tilde{G}_{i,j}^2 = \mathrm{Tr}(\tilde{G}^\top \tilde{G}). \tag{111}$$

From the definition of $G$ and $\tilde{G}$, we have $G_{ij} \leq 1$ and $\tilde{G}_{ij} \leq 1$. Thus, we have

$$\mathrm{Tr}(\tilde{G}^\top \tilde{G}) \leq N^2. \tag{112}$$

This means that $\mathrm{Tr}(\frac{\tilde{G}}{N}^\top \frac{\tilde{G}}{N}) \leq 1$. Since $G$ is the positive definite matrix, and from the above trace inequality, all the eigenvalues of $\frac{\tilde{G}}{N}$ is smaller than 1. Thus, from the trace inequality, we have

$$1 \geq \mathrm{Tr}(\tilde{G}^\top \tilde{G})/N^2 \geq \det(\tilde{G}^\top \tilde{G})^{1/N}/N, \tag{113}$$

by taking the log, we have

$$0 \geq \ln \mathrm{Tr}(\tilde{G}^\top \tilde{G})/N^2 \geq \frac{1}{N}\ln \det(\tilde{G}^\top \tilde{G}) - \ln N. \tag{114}$$

Thus we have

$$\ln \mathbb{E}_{\rho_{\mathrm{E}}(\theta)} p(x|\theta) \geq \mathbb{E}_{\rho_{\mathrm{E}}(\theta)} \ln p(x|\theta) \geq \mathbb{E}_{\rho_{\mathrm{E}}(\theta)} \ln p(x|\theta) + \frac{1}{N}\ln \det(\tilde{G}^\top \tilde{G}) - \ln N. \tag{115}$$

We can derive different form of the lower bound from Eq.(18):

$$\ln \mathbb{E}_{\rho_{\mathrm{E}}(\theta)} p(x|\theta) \geq \mathbb{E}_{\rho_{\mathrm{E}}(\theta)} \ln p(x|\theta) - \frac{1}{N} \sum_{i=1}^{N} \ln \sum_{j=1}^{N} \frac{G_{ij}}{N}$$

$$\geq \mathbb{E}_{\rho_{\mathrm{E}}(\theta)} \ln p(x|\theta) - \ln \sum_{i,j=1}^{N} \frac{G_{ij}}{N^2}$$

$$\geq \mathbb{E}_{\rho_{\mathrm{E}}(\theta)} \ln p(x|\theta) - \ln \frac{\mathrm{Tr}(\tilde{G}^{\top}\tilde{G})}{N^2}. \tag{116}$$

Then applying Eq.(114), we have

$$\ln \mathbb{E}_{\rho_{\mathrm{E}}(\theta)} p(x|\theta) \geq \mathbb{E}_{\rho_{\mathrm{E}}(\theta)} \ln p(x|\theta) - \ln \frac{\mathrm{Tr}(\tilde{G}^{\top}\tilde{G})}{N^2}. \tag{117}$$

Then we use the following trace inequality. For a positive definite matrix $A$, we have

$$\mathrm{Tr}(I - A^{-1}) \leq \det A. \tag{118}$$

This inequality come from the fact that for a positive value $\rho$, we have

$$1 - \frac{1}{\rho} \leq \ln \rho. \tag{119}$$

Since the eigenvalues of $\frac{\tilde{G}^{\top}\tilde{G}}{N^2}$ is smaller than 1, $(I - \frac{\tilde{G}^{\top}\tilde{G}}{N^2})$ is the positive definite matrix. Thus from the above inequality, we consider that $I - A^{-1} = \frac{\tilde{G}^{\top}\tilde{G}}{N^2}$, we have

$$-\ln \mathrm{Tr}(\frac{\tilde{G}^{\top}\tilde{G}}{N^2}) \geq -\ln \det((I - \frac{\tilde{G}^{\top}\tilde{G}}{N^2})^{-1})$$

$$\geq \ln \det(I - \frac{\tilde{G}^{\top}\tilde{G}}{N^2})$$

$$\geq \ln \det(I - \frac{\tilde{G}}{N}) + \ln \det(I + \frac{\tilde{G}}{N})$$

$$\geq \ln \det(I - \frac{\tilde{G}}{N}). \tag{120}$$

In the above inequality, we first applied Eq.(118) and used $\det A^{-1} = \frac{1}{\det A}$ in the second line.

$$\ln \mathbb{E}_{\rho_{\mathrm{E}}(\theta)} p(x|\theta) \geq \mathbb{E}_{\rho_{\mathrm{E}}(\theta)} \ln p(x|\theta) + \ln \det(I - \frac{\tilde{G}}{N}). \tag{121}$$

# E  Additional PAC-Bayesian generalization error bounds

Here, we present the PAC-Bayesian bounds, which are related to w-SGLD, GFSF, and DPP. We can derive those bounds from the results in Appendix C.3 and the second-order Jensen inequalities.

## E.1  Ensemble PAC-Bayesian bound

Using the prior distribution introduced in Appendix C.3, we get the generalization error bound for the ensemble setting. For simplicity, we define

$$R_c(x, h_w) = \frac{1}{2(2h_w)^2} \frac{1}{N^2} \sum_{i,j=1}^{N} \left( \ln p(x|\theta_i) - \ln p(x|\theta_j) \right)^2, \tag{122}$$

where

$$h_w^{-2} = \exp\left( 2 \min_i \ln p(x|\theta_i) - 2 \max_j \ln p(x|\theta_j) \right). \tag{123}$$

**Theorem 10.** *For all $x, \theta$, $p(x|\theta) < \infty$ and for any prior distribution $\pi_E$ over $\Theta_E$ independent of $\mathcal{D}$ and for any $\xi \in (0,1)$ and $c > 0$, with probability at least $1 - \xi$ over the choice of training data $\mathcal{D} \sim \nu^{\otimes D}(x)$, for all probability distributions $\rho_E$ with $\mathrm{supp}(\rho_E) \subset \Theta_E$, we have*

$$\mathrm{CE} \leq -\mathbb{E}_\nu[\mathbb{E}_{\rho_E} \ln p(x|\theta) + R_c(x, h_w)]$$

$$\leq -\frac{1}{D}\sum_{d=1}^{D}[\mathbb{E}_{\rho_E}\ln p(x_d|\theta) + R_c(x_d, h_w)] + \frac{\mathrm{KL}(\rho_E, \pi_E) + \frac{\ln \xi^{-1} + \Psi'''_{\pi,\nu}(c,D)}{2}}{cD}, \tag{124}$$

*where*

$$\Psi'''_{\pi,\nu}(c,D) := \ln \mathbb{E}_{\pi(\theta,\theta')}\mathbb{E}_{\mathcal{D}\sim\nu^{\otimes D}(x)}\mathrm{e}^{cD(-\mathbb{E}_{\nu(x)}L(x,\theta,\theta') + D^{-1}\sum_{d=1}^{D}L(x_d,\theta,\theta'))}, \tag{125}$$

*and*

$$L(x,\theta,\theta') := \ln p(x|\theta) + 2^{-1}(2h_w)^{-2}((\ln p(x|\theta))^2 - \ln p(x|\theta)\ln p(x|\theta')). \tag{126}$$

### E.2    Relation to w-SGLD

We define

$$R_w(x, G) := -\frac{1}{N}\sum_{i=1}^{N}\ln\sum_{j=1}^{N}\frac{G_{ij}}{N}, \tag{127}$$

where

$$G_{ij} := \exp\left(-8^{-1}h_w^{-2}\left(\ln p(x|\theta_i) - \ln p(x|\theta_j)\right)^2\right). \tag{128}$$

Then, from the second-order Jensen inequality, we have

$$\ln \mathbb{E}_{\rho_E(\theta)}p(x|\theta)$$
$$\geq \mathbb{E}_{\rho_E(\theta)}\ln p(x|\theta) + R_c(x, h_w)$$
$$\geq \mathbb{E}_{\rho_E(\theta)}\ln p(x|\theta) + R_w(x, G)$$
$$\geq \mathbb{E}_{\rho_E(\theta)}\ln p(x|\theta). \tag{129}$$

Finally, we apply this to the result in Appendix E.1. Then we upper-bound Eq.(124) with the above inequality, and we obtain

**Theorem 11.** *For all $x, \theta$, $p(x|\theta) < \infty$ and for any prior distribution $\pi_E$ over $\Theta_E$ independent of $\mathcal{D}$ and for any $\xi \in (0,1)$ and $c > 0$, with probability at least $1 - \xi$ over the choice of training data $\mathcal{D} \sim \nu^{\otimes D}(x)$, for all probability distributions $\rho_E$ with $\mathrm{supp}(\rho_E) \subset \Theta_E$, we have*

$$\mathrm{CE} \leq -\mathbb{E}_\nu[\mathbb{E}_{\rho_E}\ln p(x|\theta) + R_c(x, h_w)]$$

$$\leq -\frac{1}{D}\sum_{d=1}^{D}[\mathbb{E}_{\rho_E}\ln p(x_d|\theta) + R_w(x_d, G)] + \frac{\mathrm{KL}(\rho_E, \pi_E) + \frac{\ln \xi^{-1} + \Psi'''_{\pi,\nu}(c,D)}{2}}{cD}, \tag{130}$$

*where $\Psi'''$ is the same as Eq.(125).*

### E.3    Relation to DPP

We define

$$R_D(x, \tilde{G}) := \frac{2}{N}\ln\det\tilde{G} - \ln N, \tag{131}$$

where

$$\tilde{G}_{ij} := \exp\left(-(4h_w)^{-2}\left(\ln p(x|\theta_i) - \ln p(x|\theta_j)\right)^2\right). \tag{132}$$

Then, from the second-order Jensen inequality, we have

$$\ln \mathbb{E}_{\rho_E(\theta)}p(x|\theta)$$
$$\geq \mathbb{E}_{\rho_E(\theta)}\ln p(x|\theta) + R_c(x, h_w)$$
$$\geq \mathbb{E}_{\rho_E(\theta)}\ln p(x|\theta) + R_D(x, \tilde{G}). \tag{133}$$

Finally, we apply this to the result in Appendix E.1. Then we upper-bound Eq.(124) with the above inequality, and we obtain

**Theorem 12.** *For all $x, \theta$, $p(x|\theta) < \infty$ and for any prior distribution $\pi_{\mathrm{E}}$ over $\Theta_{\mathrm{E}}$ independent of $\mathcal{D}$ and for any $\xi \in (0,1)$ and $c > 0$, with probability at least $1 - \xi$ over the choice of training data $\mathcal{D} \sim \nu^{\otimes D}(x)$, for all probability distributions $\rho_E$ with $\mathrm{supp}(\rho_E) \subset \Theta_{\mathrm{E}}$, we have*

$$\mathrm{CE} \leq -\mathbb{E}_\nu[\mathbb{E}_{\rho_E} \ln p(x|\theta) + R_c(x, h_w)]$$

$$\leq -\frac{1}{D} \sum_{d=1}^{D} \left[ \mathbb{E}_{\rho_E} \ln p(x_d|\theta) + R_D(x_d, \tilde{G}) \right] + \frac{\mathrm{KL}(\rho_E, \pi_{\mathrm{E}}) + \frac{\ln \xi^{-1} + \Psi'''_{\pi,\nu}(c,D)}{2}}{cD}, \tag{134}$$

*where $\Psi'''$ is the same as Eq.(125).*

### E.4   Relation to GFSF

We define

$$R_g(x, K) := -\frac{2}{\tilde{h}N} \ln \det(\epsilon I + K) + \frac{2 \ln N}{\tilde{h}N}, \tag{135}$$

where

$$K_{ij} := \exp\left( -2^{-1} \tilde{h} \ln N (4h_w)^{-2} \left( \ln p(x|\theta_i) - \ln p(x|\theta_j) \right)^2 \right). \tag{136}$$

Then, from the second-order Jensen inequality, we have

$$\begin{aligned}
&\ln \mathbb{E}_{\rho_{\mathrm{E}}(\theta)} p(x|\theta) \\
&\geq \mathbb{E}_{\rho_{\mathrm{E}}(\theta)} \ln p(x|\theta) + R_c(x, h_w) \\
&\geq \mathbb{E}_{\rho_{\mathrm{E}}(\theta)} \ln p(x|\theta) + R_g(x, K) \\
&\geq \mathbb{E}_{\rho_{\mathrm{E}}(\theta)} \ln p(x|\theta).
\end{aligned} \tag{137}$$

Finally, we apply this to the result in Appendix E.1. Then we upper-bound Eq.(124) with the above inequality, and we obtain

**Theorem 13.** *For all $x, \theta$, $p(x|\theta) < \infty$ and for any prior distribution $\pi_{\mathrm{E}}$ over $\Theta_{\mathrm{E}}$ independent of $\mathcal{D}$ and for any $\xi \in (0,1)$ and $c > 0$, with probability at least $1 - \xi$ over the choice of training data $\mathcal{D} \sim \nu^{\otimes D}(x)$, for all probability distributions $\rho_E$ with $\mathrm{supp}(\rho_E) \subset \Theta_{\mathrm{E}}$, we have*

$$\mathrm{CE} \leq -\mathbb{E}_\nu[\mathbb{E}_{\rho_E} \ln p(x|\theta) + R_c(x, h_w)]$$

$$\leq -\frac{1}{D} \sum_{d=1}^{D} \left[ \mathbb{E}_{\rho_E} \ln p(x_d|\theta) + R_g(x_d, K) \right] + \frac{\mathrm{KL}(\rho_E, \pi_{\mathrm{E}}) + \frac{\ln \xi^{-1} + \Psi'''_{\pi,\nu}(c,D)}{2}}{cD}, \tag{138}$$

*where $\Psi'''$ is the same as Eq.(125).*

## F   Comparison of our second-order Jensen inequality in Theorem 3 and that of the previous work [19, 14]

First, we discuss the difference of our loss function based second-order Jensen inequality and those of the previous work [19, 14] in terms of the derivation. Although both Our approach and previous work use the Taylor expansion up to a second order, the usage of the mean $\mu$ is different. First, let us consider the Taylor expansion of $\log$ up to a second-order around a constant $\mu$, then there exists a constant $g$ between $y$ and $\mu$ s.t.,

$$\ln y = \ln \mu + \frac{1}{\mu}(y - \mu) - \frac{1}{2g^2}(y - \mu)^2. \tag{139}$$

In the previous work [19, 14], given a random variable $Z$, they define $\mu := \mathbb{E}Z$, and $y := Z$. Then take the expectation. Then we have

$$\mathbb{E} \ln Z = \ln \mathbb{E}Z - \int_{\mathbb{R}^+} \frac{1}{2g(z)^2}(z - \mu)^2 p_Z(z) dz. \tag{140}$$

Note that the constant term and the second-order reminder term remain in the equation. Then by setting $Z := p(x|\theta)$, we get the second-order Jensen inequality of the previous work [19, 14]. We can see that the variance of the predictive distribution naturally appears since we define $\mu := \mathbb{E}p(x|\theta)$.

On the other hand, as we had seen in Appendix D.2, to derive our second-order inequality, we define $\mu = e^{\mathbb{E}\ln Z}$. Then from the Taylor expansion, we obtain

$$0 = \frac{1}{\mu}\mathbb{E}[Z - e^{\mathbb{E}\ln Z}] - \int_{\mathbb{R}^+}\frac{1}{2g(z)^2}(z-\mu)^2 p_Z(z)dz. \tag{141}$$

Compared to Eq.(140), in our Taylor expansion Eq.(141), the first-order term and reminder term remain in the equation. This results in the difference between our second-order Jensen inequalities and those of previous works.

Next, we discuss the difference of the second-order Jensen inequalities in terms of the meaning of the repulsions. In previous work [19], the second-order Jensen inequality was proved

$$\mathbb{E}_q \ln p(x|\theta) \le \ln \mathbb{E}_q p(x|\theta) - V(x), \tag{142}$$

where

$$V(x) := (2\max_\theta p(x|\theta)^2)^{-1}\mathbb{E}_q\left[(p(x|\theta) - \mathbb{E}_q p(x|\theta))^2\right]. \tag{143}$$

Then by using Lemma 2, assuming that $x := e^{\ln p(x|\theta_i)}$ and $y := e^{\ln \mathbb{E}_q p(x|\theta)}$ and we get

$$(\ln p(x|\theta_i) - \ln \mathbb{E}_q p(x|\theta))^2 p(x|\theta_i)\mathbb{E}p(x|\theta) \le (e^{\ln p(x|\theta_i)} - e^{\ln \mathbb{E}_q p(x|\theta)})^2. \tag{144}$$

Then we get

$$\mathbb{E}_q \ln p(x|\theta)$$
$$\le \ln \mathbb{E}_q p(x|\theta) - V(x)$$
$$\le \ln \mathbb{E}_q p(x|\theta) - \frac{1}{2\max_\theta p(x|\theta)^2}\mathbb{E}_q(\ln p(x|\theta_i) - \ln \mathbb{E}_q p(x|\theta))^2 p(x|\theta_i)\mathbb{E}_q p(x|\theta). \tag{145}$$

This expression is very similar to our Theorem 3, but it is different in a sense that this is not the weighted variance since $\ln \mathbb{E}_q p(x|\theta)$ appears. We cannot transform this to the mean of the loss function which is contrary to the Jensen inequality.

Thus our bound second order Jensen inequality in Theorem 3 and that of the previous work [19] is different bound, that is, ours focuses on $\mathbb{E}\ln p(x|\theta)$ and the previous work focuses on $\mathbb{E}p(x|\theta)$. We found that it is hard to claim that which is tighter. We believe it is interesting direction to study in what problems which bound is appropriate.

We numerically found that for the regression tasks and bandit problems, our approach consistently outperform the previous work. On the other hand, for classification tasks, it seems that both methods show almost equivalent performances. See Section 5.

## G  Discussion about the repulsion force

### G.1  Transformation by the mean value theorem

As shown in the main paper, our loss repulsion can be translated to the parameter or model repulsion by using the mean value theorem. For example, there exist a parameter $\tilde{\theta}$ between $\theta_i$ and $\theta_j$ that is defined by a constant $t \in [0, 1]$ s.t. $\tilde{\theta} := t\theta_i + (1-t)\theta_j$, which satisfied

$$\ln p(x;\theta_i) - \ln p(x;\theta_j) = \frac{\partial_{\tilde{\theta}}p(x;\tilde{\theta})}{p(x;\tilde{\theta})}\cdot(\theta_i - \theta_j), \tag{146}$$

and similar relation also holds for $\ln p(y|f(x;\theta_i)) - \ln p(y|f(x;\theta_j))$. Thus, we can transform our loss repulsion to parameter or model repulsion. From

$$\|\ln p(x;\theta_i) - \ln p(x;\theta_j)\|^2 = \|\frac{\partial_{\tilde{\theta}}p(x;\tilde{\theta})}{p(x;\tilde{\theta})}\cdot(\theta_i - \theta_j)\|^2, \tag{147}$$

and neglecting the bandwidth for simplicity, we define a Gram matrix

$$K_{ij} := \exp\left(-\|\ln p(x;\theta_i) - \ln p(x;\theta_j)\|^2\right) = \exp\left(-\|\frac{\partial_{\tilde{\theta}} p(x;\tilde{\theta})}{p(x;\tilde{\theta})} \cdot (\theta_i - \theta_j)\|^2\right). \tag{148}$$

Then by taking the partial derivative with respect to $\theta_i$,

$$\partial_{\theta_i^{(d)}} K_{ij} = -2(\theta_i^{(d)} - \theta_j^{(d)}) \frac{\partial_{\tilde{\theta}^{(d)}} p(x;\tilde{\theta})}{p(x;\tilde{\theta})} K_{ij} + \text{Corr}, \tag{149}$$

where $\theta_i^{(d)}$ corresponds to the $d$-th dimension of the parameter $\theta_i$. The first term in the above corresponds to the parameter repulsion and the second term is the correction term. We can get the similar relation to the model repulsion. However, it is difficult to obtain the explicit form of $\frac{\partial_{\tilde{\theta}} p(x;\tilde{\theta})}{p(x;\tilde{\theta})}$.

### G.2 Model repulsion

**Regression**

We first discuss the relation to model repulsion. For a regression problem, we assume that $p(y|f(x;\theta))$ is the Gaussian distribution with unit variance for simplicity,

$$\ln p(y|f(x;\theta)) = -\frac{1}{2}(y - f(x;\theta))^2 + const. \tag{150}$$

For f-PVIs, assume that $K_{ij} = \exp(-\frac{1}{2h^2}\|f_i - f_j\|^2)$ where the bandwidth is $h$. Then the model repulsion is expressed as

$$\partial_{\theta_i} K(f_i, f_j) = -\frac{1}{h^2}(f_i - f_j) K_{i,j} \partial_{\theta_i} f_i. \tag{151}$$

On the other hand, the kernel function of our loss repulsion is from Eq.(17)

$$G_{ij} := \exp\left(-(8h_w^2)^{-1}\|\ln p(y|f(x;\theta_i)) - \ln p(y|f(x;\theta_j))\|^2\right)$$
$$= \exp\left(-(8h_w^2)^{-1}\frac{1}{4}\|f(x;\theta_i) - f(x;\theta_j)\|^2\|f(x;\theta_i) + f(x;\theta_j) - 2y\|^2\right). \tag{152}$$

We define $L(f_i) := \ln p(y|f(x;\theta_i))$ and $dL_{ij} := \partial_{f_i} L(f_i) + \partial_{f_j} L(f_j)$. The derivative of the Gram matrix $G$ is expressed as

$$\partial_{\theta_i} G_{ij} = -(\underbrace{(f_i - f_j)\|dL_{ij}\|^2}_{i)} + \underbrace{\partial_{f_i} L(f_i) dL_{ij}\|f_i - f_j\|^2}_{ii)})(4h_w)^{-2} G_{ij} \partial_{\theta_i} f_i. \tag{153}$$

As we discussed in the main part, the first term corresponds to the model repulsion and the second term corresponds to the correction term. Thus our loss repulsion is closely related to model repulsion.

We can further simplify the above relation as follows. We define $l(f_i, f_j) := \|f_i + f_j - 2y\|^2$. We define a constant $l_0$ as a constant that satisfies $l_0 \le \min_{(i,j)} l(f_i, f_j)$, we get

$$G_{i,j} \le \exp\left(-(8h_w^2)^{-1}\frac{l_0}{4}\|f_i - f_j)\|^2\right) := G_{i,j}^0. \tag{154}$$

Then by taking the partial derivative

$$\partial_{\theta_i} G_{i,j}^0 = -(4h_w)^{-2} l_0 (f_i - f_j) G_{i,j}^0 \partial_{\theta_i} f_i, \tag{155}$$

and thus, this repulsion force corresponds to the f-PVIs.

**Classification**

For classification task, if the class number is C, then, in standard models

$$p(x|\theta) := \text{Multinomial}(y|\text{softmax}(f(x;\theta))), \tag{156}$$

and $f$ is a C-dimensional output neural network. We assume that there is $N$ number of ensembles. We express $f_i^c := f^c(x, \theta_i)$ as the $c$-th output of the $i$-th neural network. We express the output of the softmax function as $\{p_i^{c'}\}_{c'=1}^c$. Then if the true class label is $y = t$, we have

$$\ln p(y|f(x; \theta_i)) = f_i^t - \ln \sum_c e^{f_i^c}. \tag{157}$$

First of all, we consider the model repulsion of f-PVI. Assume that $K_{ij} = \exp(-\frac{1}{2h^2}\|f_i - f_j\|^2)$ and the model repulsion of f-SVGD is expressed as

$$-\frac{1}{h^2} \sum_{c'=1}^c (f_i^{c'} - f_j^{c'})K_{ij}\partial_{\theta_i} f_i^{c'}. \tag{158}$$

Then, we directly calculate the derivative of the loss repulsion and connect it to the model repulsion. We can write the $(i, j)$-th element of the gram matrix as

$$G_{i,j} := \exp\left(-(8h_w^2)^{-1}(\ln p(y|f(x; \theta_i)) - \ln p(y|f(x; \theta_j)))^2\right)$$
$$= \exp\left(-(8h_w^2)^{-1}\|(f_i^t - f_j^t - (\ln \sum_c e^{f_i^c} - \ln \sum_c e^{f_j^c})\|^2\right). \tag{159}$$

Then by calculating the derivative of the Gram matrix, we have

$$\partial_{\theta_i} G_{ij} = -\sum_{c'=1}^c (\underbrace{(f_i^{c'} - f_j^{c'})}_{i)} - \underbrace{(Z_i^{c',t} - Z_j^{c',t})}_{ii)})(\delta_{c',t} - p_i^{c'})(2h_w)^{-2}G_{ij}\partial_{\theta_i} f_i^{c'}, \tag{160}$$

where

$$\delta_{c',t} = \begin{cases} 1 & (c' = t) \\ 0 & (c' \neq t), \end{cases} \tag{161}$$

and

$$Z_i^{c',t} = \ln \sum_{c''=1}^c e^{f_i^{c''} - f_i^{c'}\delta_{c'' \neq t}}, \tag{162}$$

where

$$\delta_{c'' \neq t} = \begin{cases} 1 & (c'' \neq t) \\ 0 & (c'' = t). \end{cases} \tag{163}$$

Thus, in Eq.(160), similary to the regression setting, the first term $i)$ corresponds to the model repulsion and the second term is the correction term.

For simplicity, we define $f_i := f(x; \theta_i)$. And we define $d_1(f_i^c, f_j^c) := |f_i^c - f_j^c|$ and $d_2(f_i, f_j) := |\ln \sum_c f_i^c - \ln \sum_c f_j^c|$.

$$G_{i,j} \leq \exp\left(-(8h_w^2)^{-1}\|f_i - f_j\|^2 + 2(8h_w^2)^{-1}d_1(f_i^c, f_j^c)d_2(f_i, f_j)\right) := \tilde{G}_{i,j}. \tag{164}$$

Moreover, we define a constant $d_1^0$ such that $d_1^0 \leq \max_{(c,i,j)} |f_i^c - f_j^c|$ and define a constant $d_2^0$ such that $d_2^0 \leq \max_{(i,j)} |\ln \sum_c f_i^c - \ln \sum_c f_j^c|$, we define

$$G_{i,j}^0 := \tilde{c}\exp\left(-(8h_w^2)^{-1}\|f_i - f_j\|^2\right), \tag{165}$$

where $\tilde{c} := \exp\left(2(8h_w^2)^{-1}d_1^0 d_2^0\right)$. These satisfies

$$G_{i,j} \leq \tilde{G}_{i,j} \leq G_{i,j}^0. \tag{166}$$

Then by taking the partial derivative

$$\partial_{\theta_i} \tilde{G}_{i,j} = -(2h_w)^{-2}(f_i^t - f_j^t)G_{i,j}\partial_{\theta_i} f_i^t. \tag{167}$$

Thus, our loss repulsion is closely related to model repulsion.

### G.3 Parameter repulsion

We found that it is difficult to derive the parameter repulsion without using the mean value theorem. On the other hand, when we include the additional distribution into the second-order Jensen inequality, we can derive the relation to the parameter repulsion. We use the following lemma:

**Lemma 4.** *For any distribution $p(\theta)$ that is bounded below $0 \leq \pi(\theta) < M$, under the same assumptions with Collorary 1, we have*

$$\ln \mathbb{E} p(x|\theta) \geq \mathbb{E} \ln[p(x|\theta)p(\theta)] + R(x,\theta) + \frac{1}{8h_\omega^2 N^2} \sum_{i,j=1}^{N} (\ln p(\theta_i) - \ln p(\theta_j))^2 - \ln M. \quad (168)$$

*where $R$ is the same as Theorem 3.*

*Proof.* From our second order Jensen inequality, we have

$$\ln M \geq \ln \mathbb{E}_{\rho_E(\theta)}[p(\theta)] \geq \mathbb{E}_{\rho_E(\theta)}[\ln p(\theta)] + \frac{1}{8h_\omega^2 N^2} \sum_{i,j=1}^{N} (\ln p(\theta_i) - \ln p(\theta_j))^2. \quad (169)$$

Then by combining this with Eq.(16), Lemma 4 is proved. $\square$

This lemma state that by introducing the distribution $p(\theta)$, we obtain the lower bound and repulsion term based on $p(\theta)$. For example, we can use the prior distribution $\pi(\theta)$ for $p(\theta)$.

As we did in the main paper, we can lower bound the variance of $\ln p(\theta)$ in various ways. And assume that we lower bound it in gram matrix form

$$K_{i,j} := \exp\left(-(8h_\omega^2)^{-1}(\ln p(\theta_i) - \ln p(\theta_j))^2)\right). \quad (170)$$

Assume that $p(\theta)$ is a exponential family distribution,

$$\ln p(\theta) = \eta u(\theta) + Const, \quad (171)$$

where $\eta$ is a natural parameter and $u(\theta)$ is a sufficient statistics. Then we have

$$K_{i,j} := \exp\left(-(8h_\omega^2)^{-1}\eta^2(u(\theta_i) - u(\theta_j))^2)\right). \quad (172)$$

Then by taking the partial derivative, we have

$$\partial_{\theta_i} K_{i,j} = -(8h_\omega^2)^{-1}\eta^2(u(\theta_i) - u(\theta_j))K_{i,j}\partial_{\theta_i} u(\theta_i). \quad (173)$$

In standard PVIs, the parameter repulsion force is

$$\partial_{\theta_i} K_{i,j} = -(8h_\omega^2)^{-1}(\theta_i - \theta_j)K_{i,j}. \quad (174)$$

In order to discuss these repulsions, we assume that $p(\theta)$ is a standard Gaussian distribution. Then by taking the partial derivative, we have

$$\partial_{\theta_i} K_{i,j} = -h_\omega^2(\theta_i - \theta_j)d_4(i,j)K_{i,j} - h_\omega^2(\theta_i - \theta_j)^2\partial_{\theta_i}d_4(i,j)K_{i,j}. \quad (175)$$

where $d_4(i,j) := |\theta_i + \theta_j|^2$.

On the other hand, and if there exists a constant $d_4^0$ such that $d_4^0 \leq \min_{i,j} d_4(i,j)$. Then

$$K_{ij} \leq K_{i,j}^0 := \exp\left(-h^2 d_4^0(\theta_i - \theta_j)^2)\right). \quad (176)$$

and by taking the partial derivative, we have

$$\partial_{\theta_i} K_{i,j}^0 = -h_\omega^2(\theta_i - \theta_j)d_4^0(i,j)K_{i,j}^0. \quad (177)$$

Note that the prior repulsion force is the same as the repulsion force of PVIs of Eq.(174).

### G.4 Data summation inside the variance

In the existing model repulsion force, for example, regression tasks, the kernel function $K$ is defined as

$$K_{ij} := \exp(-\|f_i(\boldsymbol{x}) - f_j(\boldsymbol{x})\|^2) = \exp(-(\|f_i(x_1) - f_j(x_1)\|^2 + \ldots + \|f_i(x_b) - f_j(x_b)\|^2)), \tag{178}$$

where $b$ is the minibatch size. Thus, we take the summation with respect to the data points.

Compared to this model repulsion, our loss repulsion is, for example, expressed as

$$-\frac{1}{D} \sum_{d=1}^{D} \frac{1}{2(2h_\omega)^2 N^2} \sum_{ij} \|\ln p(x_d|\theta_i) - \ln p(x_d|\theta_j)\|^2. \tag{179}$$

As defined in Section 3.2, when we define our Gram matrix $G$, we can incorporate the data summation term inside the Gram matrix as

$$G_{ij} := \exp\left(-(8h_w^2)^{-1} \sum_{d=1}^{D} (\ln p(x_d|\theta_i) - \ln p(x_d|\theta_j))^2\right). \tag{180}$$

Applying the Jensen inequality, we obtain

$$\frac{1}{D} \sum_{d=1}^{D} \ln \mathbb{E}_{\rho_E(\theta)} p(x_d|\theta) \geq \frac{1}{D} \sum_{d=1}^{D} \mathbb{E}_{\rho_E(\theta)} \ln p(x_d|\theta) - \frac{1}{DN} \sum_{i=1}^{N} \ln \sum_{j=1}^{N} \frac{G_{ij}}{N} \geq \frac{1}{D} \sum_{d=1}^{D} \mathbb{E}_{\rho_E(\theta)} \ln p(x_d|\theta). \tag{181}$$

## H  Relation to the misspecified model setting

In the previous work [19], the advantage of the second-order Jensen inequality was analyzed in the case of the misspecified model setting, that is, for any $\theta$, $\nu(x) \neq p(x|\theta)$. They proved the following theorems:

**Theorem 14.** *[19] Let us denote $\theta_{\mathrm{ML}}^* := \arg\min_\theta \mathrm{KL}(\nu(x), p(x|\theta))$ and $p_{\mathrm{ML}}$ is the distribution obtained by minimizing $\mathbb{E}_{p(\theta),\nu(x)}[-\ln p(x|\theta)]$. Then $p_{\mathrm{ML}}$ also minimizes $\mathrm{CE}(p) := \mathbb{E}_{\nu(x)}[-\ln \mathbb{E}_{p(\theta)} p(x|\theta)]$ if and only if for any distribution $p$ over $\Theta$ we have that*

$$\mathrm{KL}(\nu(x), p(x|\theta_{\mathrm{ML}}^*)) \leq \mathrm{KL}(\nu(x), \mathbb{E}_p p(x|\theta)). \tag{182}$$

*and $p_{\mathrm{ML}}$ can always be characterized as a Dirac distribution center around $\theta_{\mathrm{ML}}^*$, tha is, $p_{\mathrm{ML}} = \delta_{\theta_{\mathrm{ML}}^*}(\theta)$.*

According to this theorem, the previous work [19] claimed that Bayesian posterior distribution is an optimal strategy under perfect model speccification since $\mathrm{KL}(\nu(x), p(x|\theta_{\mathrm{ML}}^*)) = 0$ and $p_{\mathrm{ML}}^*$ minimizes $\mathrm{CE}(p)$.

However, in many practical settings, we cannot expect the perfect model specification. Then under the misspecified model settings, the previous work [19] clarified that the second order Jensen inequality provides the better solution as follows:

**Theorem 15.** *[19] Let us denote the $p_V^*$ as the distribution obtained by minimizing $\mathbb{E}_{p,\nu(x)}[-\ln p(x|\theta)] - \mathbb{E}_{\nu(x)} V(x)$ and $p_{\mathrm{ML}}$ is the distribution obtained by minimizing $\mathbb{E}_{p,\nu(x)}[-\ln p(x|\theta)]$. Then following inequality holds,*

$$\mathrm{KL}(\nu(x), \mathbb{E}_{p_V^*} p(x|\theta)) \leq \mathrm{KL}(\nu(x), \mathbb{E}_{p_{\mathrm{ML}}} p(x|\theta)). \tag{183}$$

*Here the equality holds if we are under perfect model specification, that is, there exists a parameter $\theta^*$ that satisfies $\nu(x) = p(x|\theta^*)$.*

Thus, this theorem clarifies that under model misspecified setting, learning the second-order Jensen inequality can be a better strategy than Bayesian inference.

Motivated these previous results, we can show the similar inequality for our loss function based second-order Jensen inequality:

**Theorem 16.** *Let us denote the $p_R^*$ as the distribution obtained by minimizing $\mathbb{E}_{p,\nu(x)}[-\ln p(x|\theta)] - \mathbb{E}_{\nu(x)}R(x)$ and $p_{\mathrm{ML}}$ is the distribution obtained by minimizing $\mathbb{E}_{p,\nu(x)}[-\ln p(x|\theta)]$. Then following inequality holds,*

$$\mathrm{KL}(\nu(x), \mathbb{E}_{p_R^*}p(x|\theta)) \leq \mathrm{KL}(\nu(x), \mathbb{E}_{p_{\mathrm{ML}}}p(x|\theta)). \tag{184}$$

*and the equality holds if we are under perfect model specification.*

The proof of this theorem is exactly the same as that of Theorem 15[19]. Here, we show the outline of the proof.

*Proof.* Define $\Omega$ as the space of distributions $p$ over $\Theta$ that satisfies $\mathbb{E}_{\nu(x)}R(x) = 0$. Then we have

$$\min_{p \in \Omega} \mathbb{E}_{p,\nu(x)}[-\ln p(x|\theta)] - \mathbb{E}_{\nu(x)}R(x) = \min_{p \in \Omega} \mathbb{E}_{p,\nu(x)}[-\ln p(x|\theta)] = \mathbb{E}_{p_{\mathrm{ML}},\nu(x)}[-\ln p(x|\theta)], \tag{185}$$

where the last equality comes from Lemma A.6 [19] (just replace V(x) with R(x)). Then we have

$$\mathbb{E}_{p_R^*,\nu(x)}[-\ln p(x|\theta)] - \mathbb{E}_{\nu(x)}R(x) \leq \mathbb{E}_{p_{\mathrm{ML}},\nu(x)}[-\ln p(x|\theta)]. \tag{186}$$

Here the left-hand side is the minimum of the second order Jensen inequality for all the distributions $p$ over $\Theta$ and right-hand is the minimum within $\Omega$. Then from the second-order Jensen inequality, we have

$$\mathrm{CE}(p_R^*) \leq \mathbb{E}_{p_{\mathrm{ML}},\nu(x)}[-\ln p(x|\theta)]. \tag{187}$$

From lemma A.6 [19],

$$\mathbb{E}_{p_{\mathrm{ML}},\nu(x)}[-\ln p(x|\theta)] = \mathrm{CE}(p_{\mathrm{ML}}), \tag{188}$$

thus we get

$$\mathrm{CE}(p_R^*) \leq \mathrm{CE}(p_{\mathrm{ML}}). \tag{189}$$

By adding the entropy of $\nu(x)$, we get the inequality of the Theorem.

Next we study when the equality holds. From Theorem 14[19], under the perfect model specification, we have that for any $p$ over $\Theta$, we have

$$\mathrm{CE}(p_{\mathrm{ML}}) \leq \mathrm{CE}(p). \tag{190}$$

Combining this with Eq.(189), we obtain

$$\mathrm{CE}(p_R^*) = \mathrm{CE}(p_{\mathrm{ML}}), \tag{191}$$

under perfect model specification. $\square$

# I  Numerical experiments

In this section, we describe the detail settings of the experiments. We also present the additional experimental results. We used two types of $h$, one is defined in Eq.(13) and the other is $h_m$ in Theorem 4.

## I.1  Toy data experiments

### I.1.1  Detail settings of the main paper

The setting is the same as that of the previous work of f-PVI [28]. We generated the data by $y = x + \sin 4(x + \epsilon) + \sin 13(x + \epsilon) + \epsilon, \epsilon \sim N(0, 0.0009)$. We generated $x$ as follows: 12 points are drawn from Uniform$(0, 0.6)$ and 8 points from Uniform$(0, 0.8)$. We used the Adam optimizer with a learning rate of 0.001. We fixed the observation variance $N(y|f(x;\theta))$ during the optimization with 0.2. In addition to the main paper, here, we also show the result of VAR-svgd and GFSF.

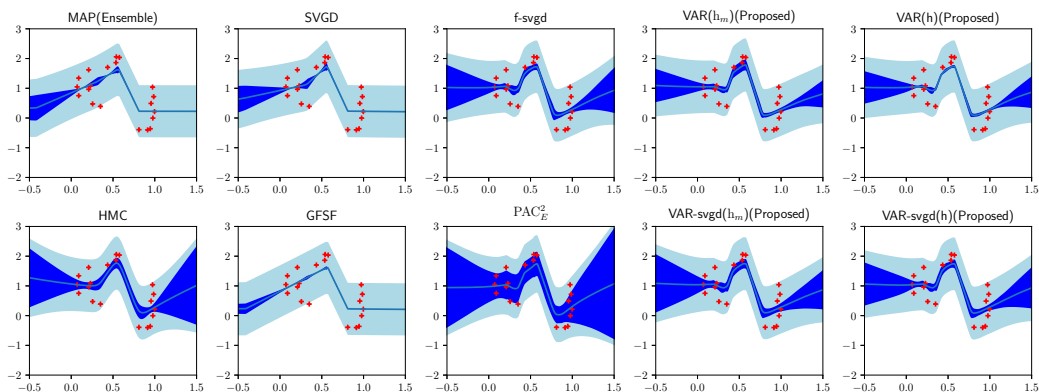

Figure 2: Uncertainty of the regressions. Blue line is the predictive mean, dark shaded area is the epistemic uncertainty from $\rho_E(\theta)$, and light shaded area is aleatory uncertainty that comes from $p(x|\theta)$.

Table 6: Comparison of performances

| | | Test Accuracy | | | | | | Test log likelihood | | | |
|---|---|---|---|---|---|---|---|---|---|---|---|
| HMC | MAP | f-SVGD | $\mathrm{PAC}_E^2$ | VAR(h) | VAR-svgd(h) | HMC | MAP | f-SVGD | $\mathrm{PAC}_E^2$ | VAR(h) | VAR-svgd(h) |
| 5.089 | 5.079 | 5.079 | 5.220 | 5.075 | 5.070 | 3.290 | 3.171 | 3.171 | 3.310 | 3.180 | 3.190 |

### I.1.2 Additional toy data experiments of the regression task

Here, we consider the linear regression problem for toy data experiment, mainly focusing on the model misspecified setting discussed in Appendix H.

We generated the toy data following $y = x + 1 + \epsilon$ where $\epsilon \sim N(0,5)$ and $x \sim \mathrm{Uniform}(-10,10)$ and thus this is the one-dimensional regression task. As a model, we prepared $y = \theta_1 x + \theta_2 + \epsilon'$ where $\epsilon \sim N(0,1)$. We used the standard Gaussian priors for each $\theta$s. Thus, this is a model misspecified setting discussed in Appendix H. We used 10 particles and optimized them in the framework of MAP, SVGD, $\mathrm{PAC}_E^2$ and our proposed approach. We optimized each model by Adam with stepsize 0.001. We also show the result of the HMC, which is the baseline method in Bayesian inference. The result is shown in Figure 3, which visualizes $95\%$ credible intervals for the prediction and mean estimate corresponding to aleatoric and epistemic uncertainties. As shown in the figure, SVGD shows almost similar uncertainty as HMC. Note that HMC visualizes the uncertainty of Bayesian inference. On the other hand, the second-order method of ours and $\mathrm{PAC}_E^2$ show larger epistemic uncertainty than those of HMC and SVGD. Note that in the result of $\mathrm{PAC}_E^2$, most training data points are inside the $95\%$ credible intervals. In Table 6, we compared the quality of fittings of the models. As we can see, except for $\mathrm{PAC}_E^2$, the fitting qualities are almost equivalent. Thus, as we discussed in the main paper, there is a trade-off between model fitting and enhancing diversity in ensemble learning in the second-order Jensen inequality. The results of HMC, f-SVGD seem small diversity since they are based on Bayesian inference, while $\mathrm{PAC}_E^2$ shows sufficient diversity. On the other hand, the quality of model fitting of Bayesian inference seems superior to that of $\mathrm{PAC}_E^2$. It seems that our proposed VAR and VAR-SVGD seem the intermediate performance and diversity between Bayesian inference and $\mathrm{PAC}_E^2$.

### I.2 Regression

The setting is the same as that of the previous work of f-PVI [28]. We used the Adam optimizer with learning rate 0.004. We used a batch size of 100 and ran 500 epochs for the dataset size $D$ is smaller than 1000. For a larger dataset, we used a batch size of 1000 and ran 3000 epochs. The result in the table is the ten repetitions except for Protein data which is the result of 5 repetitions. In addition to the main paper, here, we also show the results of VAR-SVGD in Table 7,8.

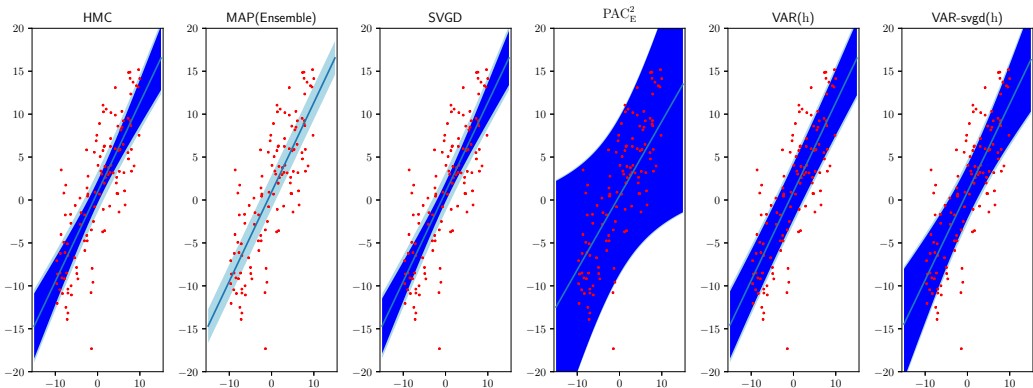

Figure 3: Uncertainty of the regressions. Blue line is the predictive mean, dark shaded area is the epistemic uncertainty from $\rho_E(\theta)$, and light shaded area is aleatory uncertainty that comes from $p(x|\theta)$.

Table 7: Benchmark results on test RMSE for the regression task

| Dataset | Avg. Test RMSE | | | | | | | |
|---|---|---|---|---|---|---|---|---|
| | MAP | SVGD | $PAC_E^2$ | f-SVGD | VAR(h) | VAR-svgd(h) | VAR($h_m$) | VAR-svgd($h_m$) |
| Concrete | 5.19±0.3 | 5.21±0.4 | 5.49±0.3 | 4.32±0.1 | 4.33±0.1 | 4.35±0.2 | 4.36±0.2 | 4.27±0.4 |
| Boston | 2.98±0.4 | 2.71±0.6 | 4.03±0.5 | 2.54±0.3 | 2.54±0.3 | 2.52±0.3 | 2.52±0.3 | 2.53±0.4 |
| Wine | 0.65±0.04 | 0.63±0.03 | 1.03±0.09 | 0.61±0.03 | 0.61±0.03 | 0.61±0.03 | 0.61±0.03 | 0.61±0.03 |
| Power | 3.94±0.03 | 3.90±0.14 | 5.04±0.21 | 3.77±0.03 | 3.76±0.03 | 3.40±0.05 | 3.76±0.06 | 3.75±0.08 |
| Yacht | 0.86±0.05 | 0.83±0.10 | 0.70±0.21 | 0.59±0.09 | 0.59±0.09 | 0.58±0.12 | 0.59±0.09 | 0.59±0.10 |
| Protein | 4.61±0.02 | 4.22±0.09 | 4.17±0.05 | 3.98±0.03 | 3.95±0.05 | 3.93±0.07 | 3.96±0.06 | 3.93±0.04 |

Table 8: Benchmark results on test negative log likelihood for the regression task

| Dataset | Avg. Test negative log likelihood | | | | | | | |
|---|---|---|---|---|---|---|---|---|
| | MAP | SVGD | f-SVGD | $PAC_E^2$ | VAR(h) | VAR-svgd(h) | VAR($h_m$) | VAR-svgd($h_m$) |
| Concrete | 3.11±0.12 | 3.11±0.14 | 3.16±0.10 | 2.86±0.02 | 2.82±0.09 | 2.80±0.06 | 2.87±0.09 | 2.81±0.06 |
| Boston | 2.62±0.2 | 2.49±0.4 | 2.61±0.3 | 2.46±0.1 | 2.39±0.2 | 2.35±0.2 | 2.48±0.4 | 2.41±0.2 |
| Wine | 0.97±0.07 | 0.96±0.06 | 1.26±0.02 | 0.90±0.05 | 0.89±0.04 | 0.89±0.06 | 0.89±0.04 | 0.90±0.07 |
| Power | 2.79±0.05 | 2.78±0.03 | 3.17±0.01 | 2.76±0.05 | 2.79±0.03 | 2.79±0.03 | 2.76±0.02 | 2.76±0.03 |
| Yacht | 1.23±0.05 | 1.32±0.6 | 0.80±0.4 | 0.96±0.3 | 0.87±0.3 | 0.81±0.2 | 1.03±0.3 | 0.91±0.2 |
| Protein | 2.95±0.00 | 2.86±0.02 | 2.84±0.01 | 2.80±0.01 | 2.81±0.01 | 2.80±0.01 | 2.80±0.01 | 2.80±0.01 |

Table 9: Benchmark results on test accuracy and negative log likelihood for the classification task

| Dataset | Test Accuracy | | | | | | | Test log likelihood | | | | | | |
|---|---|---|---|---|---|---|---|---|---|---|---|---|---|---|
| | MAP | $PAC_E^2$ | f-SVGD | VAR(h) | VAR-svgd(h) | VAR($h_m$) | VAR-svgd($h_m$) | MAP | $PAC_E^2$ | f-SVGD | VAR(h) | VAR-svgd(h) | VAR($h_m$) | VAR-svgd($h_m$) |
| MNIST | 0.981 | 0.986 | 0.987 | 0.988 | 0.988 | 0.988 | 0.988 | 0.057 | 0.042 | 0.043 | 0.040 | 0.041 | 0.041 | 0.041 |
| CIFAR 10 | 0.935 | 0.919 | 0.927 | 0.929 | 0.928 | 0.927 | 0.924 | 0.215 | 0.270 | 0.241 | 0.238 | 0.240 | 0.242 | 0.242 |

Table 10: Cumulative regret relative to that of the Uniform sampling.

| Dataset | MAP | $PAC_E^2$ | f-SVGD | VAR(h) | VAR-svgd(h) | VAR($h_m$) | VAR-svgd($h_m$) |
|---|---|---|---|---|---|---|---|
| Mushroom | 0.129±0.098 | 0.037±0.012 | 0.043±0.009 | **0.029±0.010** | 0.037±0.012 | 0.036±0.012 | 0.038±0.010 |
| Financial | 0.791±0.299 | 0.189±0.025 | 0.154±0.017 | 0.155±0.024 | 0.176±0.023 | **0.128±0.017** | 0.153±0.020 |
| Statlog | 0.675 ±0.287 | 0.032±0.0025 | 0.010±0.0003 | **0.006±0.0003** | 0.007±0.0004 | 0.008±0.0005 | 0.011±0.004 |
| CoverType | 0.610±0.051 | 0.396±0.006 | 0.372±0.007 | **0.289±0.003** | 0.320±0.005 | 0.343±0.002 | 0.369±0.004 |

## I.3    Classification

In addition to the main paper, we show the robustness to the adversarial samples in Figure 4. The experimental settings are exactly the same as that of the previous work [28].

We optimized the parameters using Adam with stepsize 0.0005. We used a batch size of 1000 and ran 1000 epochs. For MNIST adversarial experiments, we used a feed-forward network with ReLu activation and three hidden layers with 1000 units. The hyperparameter settings are the same as the result of the main paper. To generate attack samples, we used the iterative fast gradient sign method (I-FGSM) to generate the attack samples. We restrict the update with $l^\infty$ norm of the perturbation step 0.01.

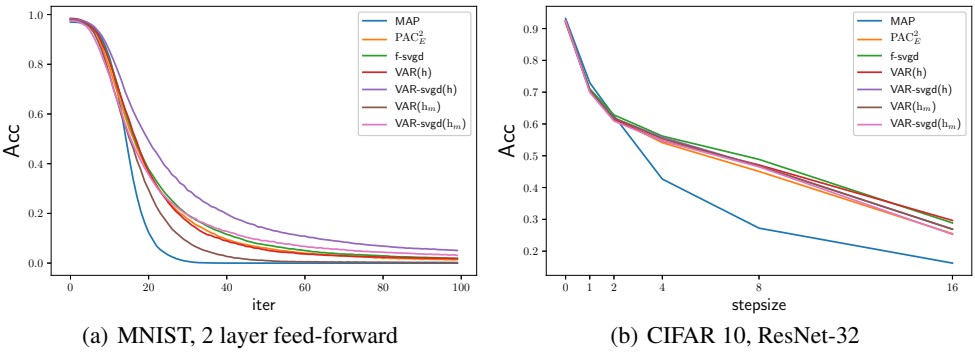

(a) MNIST, 2 layer feed-forward  (b) CIFAR 10, ResNet-32

Figure 4: Out of distribution performances

For CIFAR 10 experiments, we used ResNet32 and optimized Momentum sgd with the stepsize 0.09. We used a batch size of 128 and run 200 epochs. We generated attack samples using FGSM under different stepsizes.

The result is shown in Figure 4. For both experiments, f-SVGD, $\text{PAC}_{\text{E}}^2$, and our proposed methods showed more robustness against adversarial samples than that of the MAP estimate.

In addition to the main paper, here, we also show the results of Var-svgd in Table 9.

### I.4 Contextual bandit

First, we describe the problem setting. First we are given a context set $\mathcal{S}$. For each time step, $t = 1, \ldots, T$, a context $s_t \in \mathcal{S}$ is provided to a agent from the environment. Then, the agent choose the action $a_t$ from the available set $a_t \in \{1, \ldots, A\}$ based on the context $s_t$ and get a reward $r_{a_t,t}$. The goal of contextual bandit problem is to minimize the pseudo regret

$$R_T = \max_{g:\mathcal{S}\to\{1,\ldots,A\}} \mathbb{E}\left[\sum_{t=1}^{T} r_{g(s_t),t} - \sum_{t=1}^{T} r_{a_t,t}\right], \tag{192}$$

where $g$ denotes a mapping from the context set to available actions. For contextual bandits with non-adversarial rewards, Thompson sampling is a classical algorithm that achieves the state-of-the-art performance in practice [28]. We express the true reward generating distribution of context $s$ and action $a_t$ as $\nu_{s,a_t}$. We place a prior $\mu_{s,i,0}$ for a reward of context $s$ and action $i$. Then, this prior is updated to a posterior distribution. At each time step, Thompson sampling selects the action by

$$r_t \in \argmax_{i=\{1,\ldots,K\}} \hat{r}_{i,t}, \quad \hat{r}_{i,t} \sim \mu_{s,i,t}. \tag{193}$$

Then corresponding posterior is updated by the observed reward.

Following the previous work [28, 24], we consider a neural network where the input is the context and the output is the $K$-dimensional, and we consider a prior on parameters of the network. We approximate the posterior of the neural network and express the uncertainty by the approximate posterior distribution. All the hyperparameters are exactly the same as the previous work [28].

In addition to the main paper, here, we also show the results of VAR-SVGD in Table 10.

### I.5 Unsupervised tasks

We trained variational autoencoders (VAEs) using our theory. We trained VAEs using two types of objective functions. We express a latent variable as $z$. We use a multi-sample bound and express $z_n$ as the $n$-th random variable drawn from a posterior distribution $q(z)$. We define $l_n = p(x|z_n)\pi(z_n)/q(z_n)$. The standard objective function is a evidence lower bound (ELBO), given as $\frac{1}{N}\sum_{n=1}^{N}\ln l_n$. To obtain ELBO, we use the Jensen inequality to the marginal likelihood. We applied our second-order Jensen inequality and obtained the objective $\frac{1}{N}\sum_{n=1}^{N}\ln l_n + \frac{1}{N}\sum_{n=1}^{N}h_n(l_n - \frac{1}{N}\sum_{n=1}^{N}\ln l_n)^2$, where $h_n$ is the weights for $l_n$. This is the first objective function and we call this approach VAR.

Table 11: MNIST VAE results (N=10)

| Method | VAR-SVGD | SVGD | VAR | ELBO |
|---|---|---|---|---|
| Test LL | -88.7 | -88.6 | -89.7 | -89.9 |

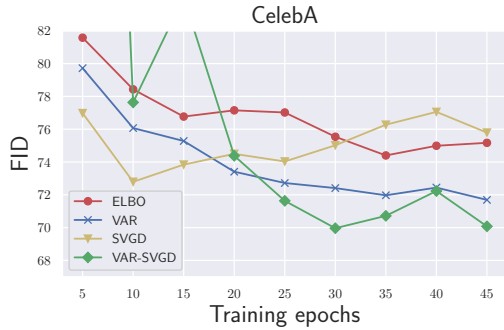

Figure 5: FID score results of CelebA dataset

Next objective function is that we use the implicit distribution for the posterior distribution in the same way as [25, 13]. We assume that $z_n$ are generated from $z_n \sim f(x, \epsilon; \phi)$ where $f$ is a deep neural network parameterized by $\phi$. A random noise $\epsilon$ is transformed to $z$ based on the input $x$. We applied the smoothing of the gradient by SVGD to approximate $\nabla_\phi \ln q(z_n)$. Thus, the gradient for the variational parameter $x$ is $\left( \sum_n k(z, z_n) \nabla_{z_n} w_n + \nabla_{z_n} k(z, z_n) \right) \nabla_\phi f(x, \epsilon; \phi)$ where $w_n = \ln p(x, z_n)$ and $k$ is the kernel function. We used the Gaussian kernel. We then incorporated the variance term as $\left( \sum_n k(z, z_n) \nabla_{z_n} \left( w_n + \frac{1}{N} \sum_n h_n (w_n - \sum_n w_n)^2 \right) + \nabla_{z_n} k(z, z_n) \right) \nabla_\phi f(x, \epsilon; \phi)$. We call this approach VAR-SVGD.

Other experimental settings, including network architectures and hyperparameters, are the same as in [25]. We used MNIST and the CelebA datasets. For MNIST, we evaluated the test log-likelihood, and we used two hidden ReLU layers with 500 units, and the latent space dimension is 8. The test likelihood was calculated by using the annealed importance sampling. The results are shown in Table 11. ELBO indicates the result obtained by optimizing the ELBO. We found that compared to ELBO, using the second-order Jensen inequality improves the performance. We found that SVGD and VAR-SVGD show almost the same performances.

For the CelebA dataset, we evaluated the FID score [10] between the real data and the randomly generated data from the models. A smaller FID score means that the distribution of the generated images is closer to the data. We used DCGAN structure for the decoder, and the latent dimension is 32. As for the encoder, we used the symmetric structure to decoder except for the final layer, which is flattened and added Gaussian noise. We used $N = 10$. The results are shown in Fig. 5. We found that incorporating the variance term in the objective function consistently improves the performance. We conjectured that since our approach can control the trade-off of the model fitting and diversity enhancing, our method can produce more diverse samples than the existing methods, which improves the performance in FID.

# J Summary of the loss function based second-order Jensen inequalities

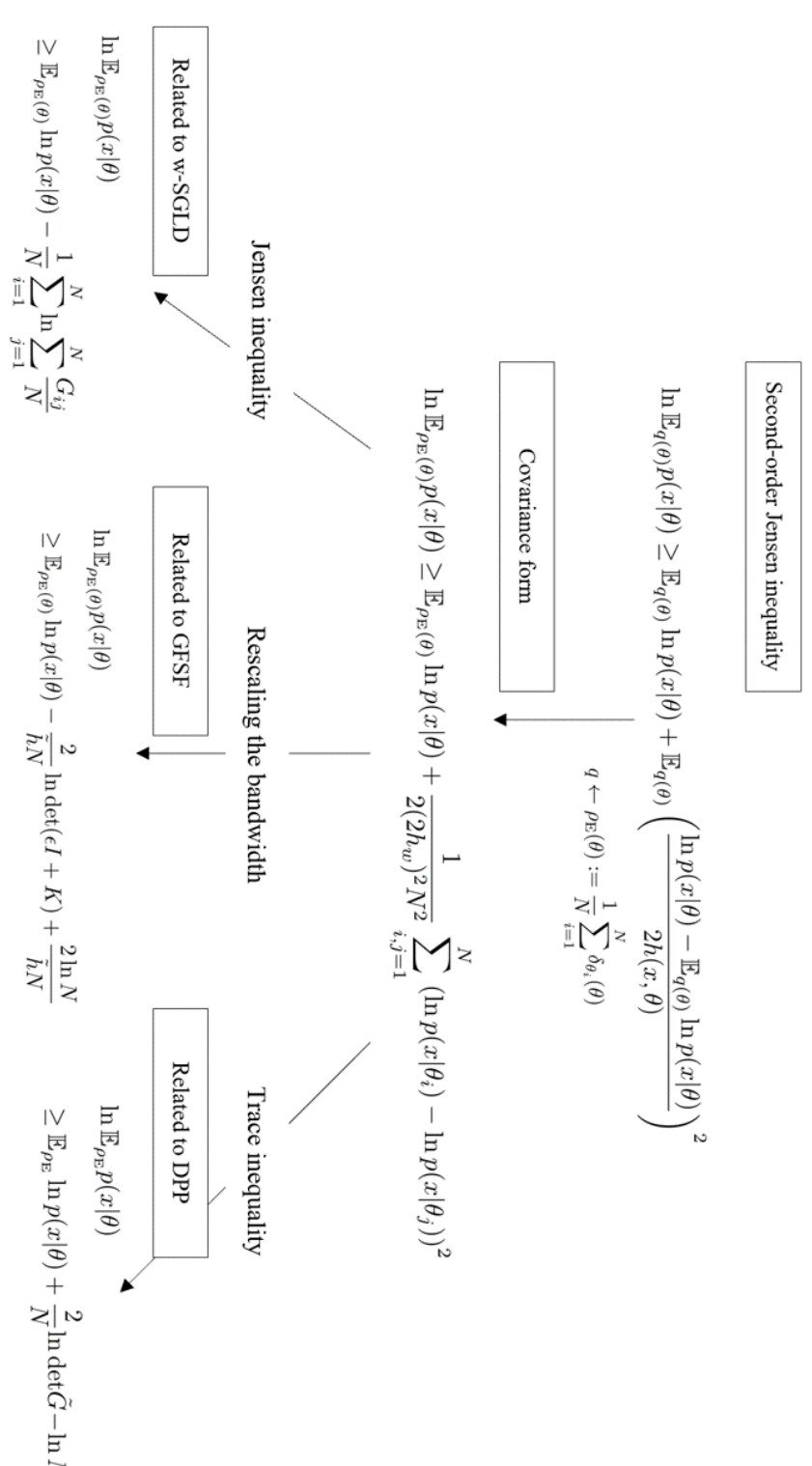

Figure 6: Summary of the second-order Jensen inequalities presented in this work.