# OpenReview forum: "Loss function based second-order Jensen inequality and its application to particle variational inference"
_NeurIPS.cc/2021/Conference — NeurIPS 2021 Poster_

### Official Review · Reviewer_LgZb · 2021-07-08

**Rating:** 7
**Confidence:** 4

**Summary:**

This paper studies particle variational inference (PVI), which is a mix between ensemble and VI (the variational posterior is assumed to be a mixture of Dirac distributions, and an additional "repulsion term" is added to give better diversity), and proposes a novel generalization to Jensen's inequality. From here, the authors propose an upper bound to the first term of the ELBO (the expected log-likelihood term), which is tighter compared to the standard Jensen inequality. Empirical findings show that the proposed loss performs well in various cases.

**Limitations And Societal Impact:**

The authors discussed both limitations and societal impacts. However, they do so in the appendix. The limitations should instead be discussed in the main text.

**Main Review:**

This paper is very well written. The proposed method is soundly and rigorously justified. I think this is a good and solid paper.

Nevertheless, one criticism from me is that the authors did not discuss the comparison of their method (i.e. their loss in eq. 25) with the standard ELBO objective. From eq. 25 it is clear that without the term $R(x_d, h)$, this objective reverts back to the standard ELBO with $q(\theta) := 1/K \sum_{k=1}^K \delta(\theta_k)$. But, when the $R(x_d, h)$ term is kept in, how valid is it to consider eq. 25 as an ELBO, and thus the proposed method as a VI method? I hope to see a discussion from the authors in the rebuttal period.

**Minor comments**

* Thm. 1:
  * Do this instead: \begin{theorem}[\citeauthor, \citeyear]
  * The statement is very hard to parse. Please consider rephrasing it
  * The notation $\nu^{\otimes D}$ is undefined
* All proofs: the qed symbols should be flush-right
* L.197: What does it mean by "$q(\theta)$ is a set of parameters"
* L.212: "a kind of PVI" -> "an instance of PVI"
* Fig. 1:
  * "Blue line" -> "Blue curve"
  * (Also L.316) How do "mean estimate" differ from the predictive mean?
* L.326, among other places: "ReLu" -> "ReLU"
* Tab. 2, 3, 4: Inconsistency with the bolding. I.e. Tab. 2 and Tab. 3 also need some bold faces
* Tab. 2: "Mnist", "Cifar" -> "MNIST", "CIFAR"
* L.359: "Variacne" -> "Variance" or just "Var"


---

**Post-Rebuttal**

After reading the authors' rebuttal and other reviews, I decided to keep recommending an acceptance.

**Time Spent Reviewing:**

3

---

> ### Author Response · Authors · 2021-08-10
> **Rebuttal: Relation between our objective in Eq.(25) and standard ELBO**
>
> Thanks for your positive feedback and for enumerating the typos.
>
> ## Relation to the standard ELBO
> As you pointed out Eq.(25) is similar to the standard ELBO. The standard ELBO is a lower bound of the marginal likelihood and standard variational inference algorithms try to maximize the marginal likelihood. On the other hand, our proposed method tries to maximize the predictive likelihood (which is equivalent to minimizing the generalization error measured in the KL divergence) shown in Theorem 4. Thus, the objective function of ELBO and ours are slightly different.
>
> However, as written in lines 197 and 198, we expect that our second-order Jensen inequality can be applied to lower-bound the marginal likelihood. This extension is different from the purpose of the current work, so we leave it for future work.
>
> ## Minor issues
> We also will correct typos and include the discussion about the limitation (Appendix A) in the main paper in our revised version.

---

> > ### Comment · Reviewer_LgZb · 2021-08-13
> > **Thanks for the response**
> >
> > Thank you for your response. Regarding the interpretation of your loss, you might find this paper [1] relevant. In any case, after reading your rebuttal and other reviews, I decided to keep recommending an acceptance.
> >
> > **References**
> > [1] Jankowiak, Martin, Geoff Pleiss, and Jacob Gardner. "Parametric Gaussian process regressors." ICML 2020.

---

### Official Review · Reviewer_hCAh · 2021-07-15

**Rating:** 7
**Confidence:** 4

**Summary:**

The paper presents a tighter PAC-Bayes bound based on the second-order Jensen inequality. Remarkably, minimizing the new bound under various relaxations/approximations recovers various PVIs and imposing a DPP prior. This novel view of PVIs explains why their repulsive force helps generalization (also, this is a valuable finite-sample regime analysis of PVIs), and also inspires new algorithms for PVIs. Experiment verifies the utility and the benefits of the new methods.

**Limitations And Societal Impact:**

The authors mentioned the limitation in explaining SVGD. I think the computational burden to estimate $h$ and $h_m$ in the proposed algorithms (the second item in "Cons") may also be a limitation.

**Main Review:**

Pros:
* I much appreciate the idea and its originality. I didn't expect a theory as fundamental as the PAC-Bayes analysis could explain specific algorithms for the Bayesian posterior. It is exciting if it is possible for an analysis at this level, as PAC-Bayes is the basic theory showing the generalization of Bayesian methods while PVIs are practical algorithms realizing the Bayesian approach, so such an analysis reveals PVIs have a theoretical root and verifies the generalization benefits of PVIs from the root (remarkably, this is a finite-particle analysis).
The analysis is based on a finer development of PAC-Bayes bound, which is also novel.
* The paper reads clear and enjoying, although it involved heavy equations. The meaning of complicated math expressions are well explained. I did not find problems regarding math deduction.

Cons:
* I find my major concern is on the proposed new PVI, presented at the beginning of Sec. 5.
  - Regarding the KL term in the objective Eq. (25), it is inappropriate to directly plug-in the empirical distribution $\rho_E$. KL would be undefined (or at least, infinite) if the first distribution is not absolutely continuous w.r.t the second distribution, which is the case for this $\rho_E$ and a common choice of $\pi$ (e.g., a standard Gaussian). From another perspective, if the entropy term of $\rho_E$ in the KL is treated as a constant, then minimizing this KL would collapse all particles onto the modes of $\pi$, which is not desired for diversity. This is also the reason why PVIs cannot be derived by treating the particles as the parameters of the variational distribution and minimizing the -ELBO objective (Interestingly, Eq. (25) is in the form of an ELBO!), and why kernels are introduced for smoothing the empirical distribution [Liu et al., 2019].
  - I did not find how $h$ and $h_m$ are calculated in the algorithm. I suppose they are estimated directly using their definitions, but this then involves a likelihood-model $p(x|\theta)$ optimization for every data sample $x$ in every iteration. Will it be too expensive?
* Literature inconsistency. In Table 1, both GFSD and GFSF are originated from [Liu et al., 2019]. Also, GFSD is not the sum of SVGD and w-SGLD. It is w-SGLD without the second term.
* Minor issues.
  - Notations. In Eq. (2), $v$ should also depend on the specific particle $\theta_i$. So it'd be in the form $v(\theta_i, \{ \theta_{i'}^{old} \}_{i'=1}^N)$ (I cannot type the curly bracket). In Line 110, $p(x_d | \theta)$ may be also needed to adapt to e.g. $p(y_d | f, x_d)$. Would be better to unify notations of $\mathbb{E} Z$ and $\mathbb{E} [Z]$.
  - The paper mentioned that there is not finite-particle analysis before. But it seems that [11] showed an analysis in this case.
  - Possible typos. Line 214: there may not be "to". Line 359: "Variacne".
  - Discussion. The paper showed the connection to the median method for bandwidth selection. The authors may also consider the connection to the heat-equation bandwidth selection method by [Liu et al., 2019] based on an equivalent dynamics principle, which achieves a better performance than the median method.

[Liu et al., 2019] "Understanding and Accelerating Particle-Based Variational Inference". ICML 2019.

**Time Spent Reviewing:**

4

---

> ### Author Response · Authors · 2021-08-10
> **Rebuttal: Discussion about KL divergence in Eq.(25) and computational cost of $h$ and $h_m$**
>
> Thank you for your positive feedback. We reply major concerns below.
>
> ## KL divergence in Eq.(25)
>  As you pointed out, we need to carefully define a prior distribution for ensemble learning so that the KL divergence between $\rho_\mathrm{E}$ and $\pi$ can be defined properly. The reason that we directly use an empirical distribution as an approximate %posterior is to characterize the finite particle VI as variational methods in our theoretical analysis.
> posterior is to confirm the theoretical findings such as Theorem 4. For that purpose, following the prior work [16], which also introduced an empirical distribution as an approximate posterior distribution, we used the specific discrete prior distribution. Using such a prior distribution, KL divergence can be defined properly. The detailed discussion is shown in Appendix C.3.1.
>
> However, such a special discrete prior distribution is an unnatural choice. Thus, we also introduced smoothing for empirical distributions. That method is shown in Appendix I, Eq.(191). We call that approach *Var-svgd*. Then, we conducted numerical experiments for Var-svgd and compared it with *Var* shown in Eq.(25). Experimental results are shown in Appendix I. We found that Var (Eq.(25)) and Var-svgd (Eq.(191)) showed competitive performance for various tasks. We also found that Var is slightly better than Var-svgd in contextual bandit problems.
>
>
> ## Computational Complexity for $h(x)$ and $h_m(x)$
> To implement the bandwidth $h(x)$ and $h_m(x)$, we are required to evaluate them for every data point. In numerical experiments, since we used the stochastic gradient descent, we only needed to evaluate them on minibatches. This saved the computation cost so much. We measured the wall clock time and compared it with SVGD. We found that the wall clock time of ours is almost equivalent to SVGD. We will report this in the revised version.
>
> ## literature inconsistency
> We will show the correct equation of GFSD in Table 1 as you pointed out.
>
> ## Minor issues
> We also revise our paper based on minor issues. Especially, the heat-equation bandwidth selection seems interesting for comparison, we will add the numerical experiments about it. We found that it is difficult to directly incorporate the heat equation method into our theory because we cannot obtain an analytic form of the bandwidth using the heat-equation method.

---

> > ### Comment · Reviewer_hCAh · 2021-08-21
> > **Thanks to authors' reply**
> >
> > Thanks for clarifying the issues. I did not notice the KL issue is discussed in Appendix C.3.1 and you may link the discussion to the context around Eq. (25). I think the discussion on the computational complexity for $h$ is also expected in the paper. In all, I keep my positive score due to the overall novelty and contribution.

---

### Official Review · Reviewer_ELqV · 2021-07-20

**Rating:** 6
**Confidence:** 3

**Summary:**

This paper derives a tighter Jensen Inequality and applied it into the framework of particle variational inference. Different from previous work, it tackles this problem in light of PAC-Bayesian analysis, and provides a new second-order Jensen inequality, which has the repulsion term based on the loss function. In addition, the authors derive the existing PVI and DPPs and demonstrate that these methods work well even at a finite ensemble size, since their objective functions includes valid diversity enhancing terms to reduce the generalization error. Finally, the numeral experiments show the better performance than the most related work PAC2.

**Main Review:**

The paper is well written and the last figure in the supplementary clearly illustrates the main idea of the loss function based second-order Jensen inequalities.

The theoretical analysis is sound to me, even I didn't check the detailed proof in the supplementary materials. The idea seems to be enlighten by [16], but the methods and mathematical tools used in this paper are different. It also builds the connection to the existing methods such as w-SGLD. From the theoretical perspective, this is good paper and should be interested to the area of probabilistic inference. However, the overall contribution seems incremental based on the previous works Stein variational gradient descent (SVGD) and "Function Space Particle Optimization for Bayesian Neural Networks". Basically, it is still a simple variant of SVGD. Besides, I would like to see the performance on generative or energy models. The regression or classification tasks are too simple to see the real improvement over SVGD.

A typo Line 221: $\ln p(x|\theta_i)$ -> $\ln p(x|\theta_j)$

**Time Spent Reviewing:**

4

---

> ### Author Response · Authors · 2021-08-10
> **Rebuttal: Main purpose of our work and additional experiments**
>
> Thank you for reviewing our paper. The main purpose of this work is to understand the performance of particle variational inference (PVI) from the viewpoint of the generalization error under a finite particle setting. Based on this, the purpose of numerical experiments is to support the theoretical findings such as Theorem 4. Theorem 4 claims that it is important to control the trade-off between the model fitting and diversity enhancement in order to reduce the generalization error. To confirm this numerically, we directly minimized a generalization error bound and compared it with other PVIs including SVGD.
>
> In numerical experiments, first, we focused on the supervised tasks since supervised tasks are easier to interpret in terms of the generalization error bound, which our theory focuses on. In addition to supervised tasks, we conducted experiments on bandit tasks in Section 5.4 and checked the robustness to adversarial attack in classification in Appendix I. These tasks require the evaluation of the uncertainty and need to balance the model fitting and diversity. We found that our proposed method showed competitive performances with SVGD in supervised tasks. On the other hand, in bandit tasks and adversarial robustness tasks, ours showed better performances than SVGD.
>
> However, as you pointed out, applying our theory to generative or energy models seems appealing, so we additionally performed numerical experiments on those models. If a paper revision is allowed like ICLR, we will add an experiment, but according to the guideline, no paper revisions may be submitted during the rebuttal process, so we only report the minimum result below.
>
>
>
>
> ## Additional experiments
> ### Generative models
> We compared our proposed method with SVGD using convolutional variational autoencoder models. Let us denote a latent variable as $z$ and observed data as $x$. We used the implicit model for a decoder following [1]. The implicit decoder was  $z\sim f(x;\theta,\epsilon)$, where $f$ is a deep neural network parameterized by $\theta$, which has the same structure as DCGAN. A random noise $\epsilon$ was transformed to $z$ based on the input $x$. The detailed network setting is the same as [1]. We updated the parameters of decoders using the amortized approach introduced in [2].
>
> We used the CelebA and MNIST datasets. For both datasets, our method and SVGD showed almost the same final performances. For MNIST, both achieve -89.5 in negative log-likelihood. For CelebA, both achieve 74 in the FID score. However, in the CelebA dataset, we found that SVGD suffers from overfitting when we increase the number of training epochs. We observed that (FID score, training epochs)=(74,20), (75,25), (77,30), (79,40) for SVGD.
> On the other hand, our proposed method less suffered from overfitting. (FID score, training epochs)=(74,20), (74,25), (75,30), (76,40) for our proposed method.
>
> We conjecture that similar NLL and FID scores mean that the model fitting performance of our approach and SVGD are almost equivalent. On the other hand, since our approach can control the trade-off of the model fitting and diversity enhancing, our method is less suffered from overfitting. This is consistent with the supervised experiments in the paper.
>
> ### Energy-based models
> Next, we applied SVGD and our method to the energy-based model introduced in [3]. We replaced stochastic gradient Langevin dynamics (SGLD) in [3] with SVGD and our method and compared the final IS score and energy of the model on the CIFAR10 dataset. About the energy of the model, our method and SVGD showed 14.0 and the baseline SGLD showed 13.5 for training images. On the other hand, about the IS score, ours and SVGD showed 2.6, and baseline SGLD showed 2.9 for generated images. We conjecture that this is because SGLD used multiple Gaussian noises and can produce diverse images. This resulted in a better IS score than ours and SVGD. The similar performances of ours and SVGD show that the model fitting performances of ours and SVGD are almost equivalent.
>
> We will report the above experimental results in our revised version in detail.
>
>
> [1] Kernel implicit variational inference (ICLR2018, J Shu et al.)
>
> [2] VAE learning via Stein variational gradient descent (Neurips2017, Y Pu et al.)
>
> [3] Learning non-convergent non-persistent short-run MCMC toward energy-based model (Neurips 2019, E Nijkamp, et al.)

---

> > ### Author Response · Authors · 2021-08-30
> > **Did we answer your questions?**
> >
> > Dear reviewer ELqV
> >
> > Thank you for reviewing our paper.
> >
> > We would appreciate it if you could confirm whether our response solved your questions or there still remain concerns.
> >
> > Best

---

> > > ### Comment · Reviewer_ELqV · 2021-09-08
> > > **Thanks to the detailed response.**
> > >
> > > Thanks. The author addressed my most concerns. I will raise my score.

---

### Decision · Program_Chairs · 2021-09-27

**Decision:**

Accept (Poster)

**Comment:**

This submission uses a tighter inequality for the likelihood than Jensen's inequality by a second order expansion.  The additional terms are then interpreted as a diversity term.  A PAC-Bayesian analysis is then developed based on this inequality.  The submission initially received a mix of scores straddling the acceptance threshold.  An external expert was consulted giving a mixed opinion (see copy of comments below), while the lower scoring reviewer was convinced to revise their score upwards during the rebuttal process.  On the balance, the paper is viewed positively, but has some important issues that should be corrected prior to publication.  These relate to incorporating reviewer comments (see below), but importantly to avoid obvious technical inaccuracies such as "MAP estimation can be used to obtain samples from the posterior" (section 2, from expert opinion below).

Additional expert opinion:
"The idea is simple overall, they use a thighter inequality than Jensen For the likelihood by exploiting second order expansion. This results in an additional terms which is interpreted as a diversity term. They combine this bound with a pac-bayesian bound to define a suitable loss for sampling from posterior in a particle fashion. Then they proceed to relate this loss to existing methods in particular w-SGLD and DPP, showing that their bound can result in methods that are formally similar to these prior works I didn’t find the paper particularly nice to read, for instance **some important details raised by reviewer 2 was only discussed in the appendix without mentioning them in the main ** The experiements did not compare with very standard benchmarks such as ELBO or even SMC methods which is also a particle methods. In the text they say that they used to HMC as a baseline, but I couldn’t see the performance of HMC in the results Also some inaccuracies in section 2.1: they claim that MAP estimation can be used to obtain samples from the posterior, but of course this is wrong... this is on the negative side on the positive side: The PAC-bayesian people might be interested in such result as it relies on a tighter bound than Jensen to derive the PAC-bayesian bound. And the connection to other methods is also interesting at a high level, although it is unclear to what extend this connection can say anything specific about w-SGLD and DPP in terms of generalization bound, etc If the paper is accepted, then the remarks by Reviewer 2 should be taken into account and in general the paper should mention in the main text what is going on in the appendix"